# Rapid threat assessment in the *Drosophila* thermosensory system

Genevieve C. Jouandet[1], Michael H. Alpert[1,6], José Miguel Simões[1,6], Richard Suhendra[2], Dominic D. Frank[1,4], Joshua I. Levy[1,5], Alessia Para[1], William L. Kath[1,2,3] & Marco Gallio [1] ✉

Neurons that participate in sensory processing often display "ON" responses, i.e., fire transiently at the onset of a stimulus. ON transients are widespread, perhaps universal to sensory coding, yet their function is not always well-understood. Here, we show that ON responses in the *Drosophila* thermosensory system extrapolate the trajectory of temperature change, priming escape behavior if unsafe thermal conditions are imminent. First, we show that second-order thermosensory projection neurons (TPN-IIIs) and their Lateral Horn targets (TLHONs), display ON responses to thermal stimuli, independent of direction of change (heating or cooling) and of absolute temperature. Instead, they track the rate of temperature change, with TLHONs firing exclusively to rapid changes (>0.2 °C/s). Next, we use connectomics to track TLHONs' output to descending neurons that control walking and escape, and modeling and genetic silencing to demonstrate how ON transients can flexibly amplify aversive responses to small thermal change. Our results suggest that, across sensory systems, ON transients may represent a general mechanism to systematically anticipate and respond to salient or dangerous conditions.

A key determinant of the dynamic processing of sensory information is the extent to which a neuron will keep responding to a sustained stimulus. ON responses are characterized as transient neural activity that coincides with stimulus onset and are found both in specialized sensory neurons as well as in downstream circuits. For example, a number of mechanosensory neurons that innervate the skin respond to sudden changes in mechanical pressure and adapt very rapidly, a "stimulus-ON" response that is important for texture discrimination[1]. In the context of vision, ON responses are recorded in specialized Retinal Ganglion Cells (RGCs), two synapses downstream of the photoreceptors (the "light-ON" pathway of the visual system[2]). Here, the light-ON circuit runs in parallel to a "light-OFF" RGC pathway and, together, the two contribute to the detection of changes in light intensity and to contrast sensitivity[2,3]. Even outside of vision and touch, ON responses appear widespread (e.g.,[4–10]), perhaps representing a universal feature of sensory coding[11]. Yet it is fair to say that the significance of ON responses to sensory processing is not universally well-understood.

We have previously demonstrated the existence of ON responses in second-order neurons of the *Drosophila* thermosensory system. In flies, rapid temperature preference behavior is mediated by dedicated cold- and hot-activated temperature receptor neurons (TRNs) housed in the antenna[12]. TRN axons target the posterior antennal lobe (PAL) region of the brain, where hot and cold terminals form two distinct, adjacent glomeruli defining a simple central map for temperature representation[12]. From the PAL, a battery of differentially tuned second

[1]Department of Neurobiology, Northwestern University, Evanston, IL, USA. [2]Department of Engineering Sciences and Applied Mathematics, Northwestern University, Evanston, IL, USA. [3]National Institute for Theory and Mathematics in Biology, Northwestern University, Chicago, IL, USA. [4]Present address: Laboratory of Social Evolution and Behavior, The Rockefeller University, New York, NY, USA. [5]Present address: Department of Immunology and Microbiology, The Scripps Research Institute, La Jolla, CA, USA. [6]These authors contributed equally: Michael H. Alpert, José Miguel Simões. ✉e-mail: marco.gallio@northwestern.edu

order projection neurons (TPNs) collect information from the TRNs and target higher brain centers such as the Lateral Horn (LH) and Mushroom Body (MB; [13,14]), where temperature signals are processed to impact a variety of innate and learned behaviors. At least 10 classes of TPNs have been functionally characterized[13,14], and their responses range from those of "thermometer" cells whose firing rates scale with absolute temperature showing little to no adaptation[15], to ON/OFF responders that display only dynamic, fast-adapting responses either at the onset or offset of a given thermal transient[13].

Of particular interest, we described a select population of thermosensory projection neurons (TPN-IIIs) that display ON calcium transients at the onset of either a heating or cooling stimulus[13]. TPN-IIIs are independently driven by hot- and cold-activated TRNs of the antenna, and we termed these cells "broadly tuned" to differentiate their activation profile from that of hot-activated or cold-activated ("narrowly tuned") TPNs. We further demonstrated that normal activity in this unusual cell type is essential for temperature preference behavior[13]. Interestingly, cells responding to both heating and cooling have been recently also discovered in the mammalian thermosensory system[16,17], suggesting a potentially conserved role for broadly tuned cell types in thermosensory processing. Here, we study the electrophysiological responses of broadly tuned TPN-IIIs in the *Drosophila* thermosensory system, we identify their targets in the higher brain, the thermosensory lateral horn output neurons (TLHONs), and probe the role of the circuit they define using genetic silencing, connectomics, modeling, and high-resolution imaging of freely behaving flies. Our results uncover a full circuit, from sensory neurons of the antenna to descending neurons targeting the ventral nerve cord, that selectively tracks the rate of a temperature change, independent from the direction of change (heating or cooling) and from absolute temperature (hot or cold). If the rate of thermal change exceeds a specific threshold, TLHONs fire a burst of action potentials (an ON response). Our work demonstrates that, during heat avoidance, this signal alerts the fly that potentially dangerous conditions may be imminent and primes locomotor behavior for a rapid escape maneuver.

## Results

The existence of broadly tuned second order TPNs in the fly thermosensory system was unexpected and, while their importance for normal thermal preference has been documented[13], we still do not understand how their activity is integrated with that from other TPNs to guide the navigational decisions that underlie thermotaxis and temperature preference behavior.

To better understand TPN-III's specific role in thermosensory processing, our first goal was to produce a driver for selective targeting. TPN-IIIs are characterized by a unique anatomy: their dendrites arborize extensively within the PAL, and they extend axons through the mediolateral antennal lobe tract, targeting the anterior aspect of the mushroom body calyx, the lower edge of the lateral horn, and the posterior lateral protocerebrum ([13,14,18] and Fig. 1a). Their function has been previously characterized using two distinct Gal4 drivers with overlapping expression in TPN-IIIs (VT040053- and R22C06-Gal4,[13]). Here, we used the promoter fragments of each driver to produce split-Gal4 reagents for intersectional targeting; the resulting TPN-III split-Gal4 drives expression in TPN-III cells but in no other cells of the brain or ventral nerve cord (Fig. 1b). We note that the VT040053::R22C06 split Gal4 driver is active in ~4−5 cells/hemisphere, fewer than those labeled by each original Gal4 (~11 in VT040053- and ~7 in R22C06-Gal4, see ref. 13). Nevertheless, silencing neural responses using this reagent (by expression of the hyperpolarizing agent Kir2.1[19]) produced stark deficits in hot and cold avoidance in 2-choice tests, comparable to those reported for the parental Gal4s (Fig. 1c−e and see ref. 13); we conclude that this more selective driver is active in a sufficient number of TPN-IIIs to be informative.

Next, we used this driver to target TPN-IIIs for 2-photon guided patch clamp electrophysiology, i.e., using GFP expression as a guide (Fig. 1f−l[15]). This approach allowed us to directly demonstrate that TPN-IIIs are indeed independently driven by hot and cold receptors of the antenna (by expressing the opsin Chrimson[20] in TRNs and recording responses in TPN-IIIs; Fig. 1f−h, and see methods for details). Moreover, we used this approach to characterize TPN-IIIs' electrophysiological responses to a series of temperature stimuli. Here, we used "step" stimuli that consists of a rapid temperature change (heating or cooling at -1 °C/s) followed by stable conditions (hot or cold for ~1 min), in turn followed by a rapid return to baseline (i.e., as a result of cooling or heating back to baseline, respectively; Fig. 1i−l). Our results confirm and expand what was previously reported for this cell type using calcium imaging[13], TPN-IIIs respond to the onset of either heating or cooling, and their activity consists of a transient burst of action potentials (Fig. 1j, k) which appears largely independent from the baseline temperature (Fig. 1k, l). Once the temperature stabilizes, the firing rate of TPN-IIIs also stabilizes so that, in stable conditions, the cell's firing rate is largely independent of absolute temperature (Fig. 1k, l). This stands in contrast, for example, with the behavior of non-adapting TPN-IIs, whose firing rate scales with absolute temperature in the cold range[15].

### TLHONs are key TPN-III targets in the Lateral Horn

How does the activity of this "broadly tuned" ON responder integrate with that of other thermosensory PNs to guide thermal preference behavior? To begin answering this question, our next goal was to identify the targets of TPN-IIIs in the higher brain. We initially used photo-activated GFP (PA-GFP) -mediated circuit mapping[21], an approach in which the pre-synaptic termini of a cell of interest are labeled using a fluorescent protein as a guide, and targeted for PA-GFP photo-conversion under a two-photon microscope. Bright, photo-converted PA-GFP is initially detected in post-synaptic structures, and will eventually label post-synaptic cells by diffusion (Fig. 2a, b, and see methods for details). Our observations were later expanded using connectomics data (i.e., based on the hemibrain connectome[22], Fig. 2c and see methods and Supplementary Table 1 for details).

One of the major target regions for TPN-III axons is the ventral portion of the Lateral Horn (Fig. 2b), and two cell types stood out as potential key outputs for TPN-IIIs in this structure: a group of ~10 cells/hemisphere that appear to lack distinct axons (the interneuron LHPV2a; Fig. 2c, purple) and LHPV2g, a cell type comprising two cell bodies/hemisphere with dendrites in the ventral LH and axons in the posterior and anterior ventrolateral protocerebrum (PVLP/AVLP; Fig. 2c, black), a region that contained prominent PA-GFP labeled fibers in our mapping experiments (arrowhead in Fig. 2b). As LHPV2gs represent an output of the Lateral Horn, we hereafter refer to these cells as Thermosensory Lateral Horn Output Neurons or TLHONs.

Connectomic analysis revealed that, in the ventral LH region, TPN-IIIs target directly LHPV2a (286 synapses), which in turn are significantly inter-connected with TLHONs (Fig. 2d, and see Supplementary Table 1 for details). We also recorded numerous inter-connections between cells belonging to each cell type (Fig. 2d). Together, these results suggest a potential pathway from the transfer of thermosensory information from the hot and cold receptor of the antenna, through TPN-IIIs, LHPV2as and onto TLHONs.

Connectomic analysis also suggested additional routes through which thermosensory information may reach TLHONs: a direct pathway from a hot-specific TPN (TPN-V, Fig. 2e,f), and a number of additional indirect pathways involving neurons of the lateral horn (Fig. 2f; note that we only considered pathways that involve a single additional intermediate cell type between TRNs and TLHONs, see methods for details and Supplementary Table 1).

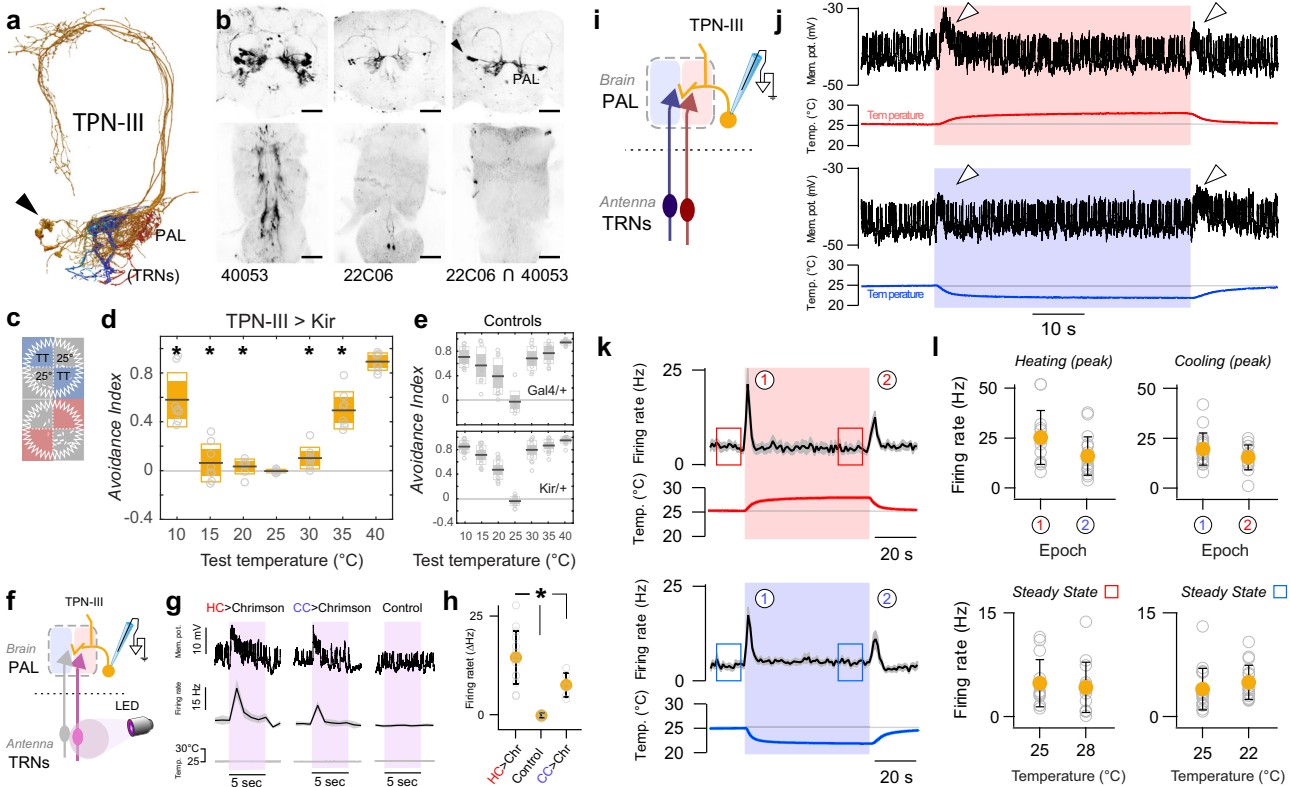

**Fig. 1 | TPN-IIIs display ON responses to temperature change. a** EM reconstruction of a group of 5 right-hemisphere TPN-IIIs (orange; arrowhead: cell bodies) and of the afferents from aristal hot and cold TRNs (red and blue, respectively) partially overlapping in the hot and cold glomeruli of the posterior antennal lobe (PAL). **b** Expression pattern of VT040053-Gal4/UAS-CD8:GFP, R22C06-Gal4/UAS-CD8:GFP and VT040053.DBD ∩ R22C06.AD/UAS-CD8:GFP; the split driver displays selective expression in TPN-IIIs (2-photon stacks of top: brain, bottom: VNC; inverted so that GFP expression is in black; arrowhead: cell bodies; ∩ = intersection; images represent 5 independent repeats; scale bars = 50 μm). **c-e** Silencing TPN-IIIs produces defects in both hot and cold avoidance in a 2-choice test for temperature preference. **c** Assay schematic. Groups of 15 flies are given a choice between 25 °C (gray shading) and a test temperature (TT; blue shading on top and red shading on the bottom), the time in each temperature is used to quantify an avoidance index (AI). **d** AIs for flies in which TPN-IIIs are silenced by expression of Kir2.1 (under the control of VT040053-Gal4.DBD ∩ R22C06-AD); * = p < 0.05, p = [0.001,2.09E−07,5.31E−05,0.717,3.42E−12,2.83E−06,0.069] for 25 °C vs 10,15,20,25,35,40 °C, respectively. **e** Control genotypes (**d** and **e**: black/red line = mean, inner box = 95% CI, outer box = SD; individual points = # of groups; N = [8,10,16] groups at each temperature for Gal4/Kir, Ga4/+, Kir/+, respectively,

groups, 20 animals/group; 2-way ANOVA). **f-k** 2-photon guided patch clamp electrophysiology of TPN-IIIs. **f-h** Optogenetic activation of hot or cold TRNs of the antenna drives firing in TPN-IIIs. **f** Experiment schematic. **g** Example TPN-III whole cell recording from flies expressing CsChrimson in hot or cold TRNs and in which TPN-IIIs are independently labeled by GFP (control lacks CsChrimson expression), pink boxes represent red light stimulation (N = 14,7,6 cells for firing rates; line and shading = mean ± SEM). **h** Quantifications of peak firing rates during light stimulation (gray dots indicate individual cells, orange circles indicate mean ± SD; N = 14,7,6 cells, * = significantly different from Control, p = 9.04E−06, 1-way ANOVA). **i-k** TPN-III display comparable transient "ON" responses to temperature change, irrespective of direction (heating or cooling) and of absolute temperature. **i** Schematic of the recording (**j**), raw traces, (**k**) average firing rate histograms, and (**l**) quantification of firing rate. Peak responses to heating (red shading) and cooling (blue shading) are comparable (circled numbers in **k** and top row plots **l**). Firing rates at stable temperatures of 22 °C, 25 °C, and 28 °C are also comparable (boxes in **k** and bottom row plots **l**; in **k** and **l** n = 15 cells/hot and 18 cells/cold from 9 animals, mean ± SEM; colored circles in (**l**) are mean ± SD). Source data are provided as a source data file.

## TLHON activity is essential for normal temperature preference behavior

Because TLHONs potentially receive broad thermosensory drive, we reasoned that they may represent a key output of the lateral horn for thermosensory behavior, and next focused our attention on this cell type. First, we identified a Gal4 driver selectively expressed in these cells. TLHON-Gal4 labels all TLHONs (Fig. 2g) and co-expression of a dendritic (DenMark[23]) and synaptic (syt-GFP[24]) marker in this cell under the control of this driver confirmed their polarity (Fig. 2h; see methods for a detailed description of the full expression pattern of this line, and further below for controls).

Next, we tested the idea that TLHONs may indeed represent a significant functional output for TPN-IIIs:

(1) "Artificial" activation of TPN-IIIs (by transgenic expression of P2X2, an exogenous ATP receptor, and ATP application[25]) resulted in calcium responses in TLHONs (independently monitored by G-CaMP[26], Fig. 2i, j).

(2) Genetic silencing of TLHONs (by transgenic expression of Kir2.1) produced striking defects in both cold and hot avoidance, very similar to those observed as a result of TPN-III silencing (Fig. 2k and compare to Fig. 1d; and see Supplementary Fig. 1 for recordings demonstrating effective silencing by Kir2.1).

(3) Genetic silencing of TLHONs also resulted in a specific deficit in the avoidance of dry air conditions (Fig. 2l), an effect that has been previously reported for TPN-IIIs and that is explained by their involvement in the processing of combined temperature/humidity stimuli[27].

Interestingly, our connectomic analysis also revealed potential TLHON drive from four olfactory PNs originating from three antennal lobe glomeruli (VL2p, VM4, VL2a; in addition to potential drive from additional olfactory glomeruli through alternative indirect LH pathways, Fig. 3a,b, and see Supplementary table 2 for IDs). Notwithstanding these connections, silencing TLHONs did not affect

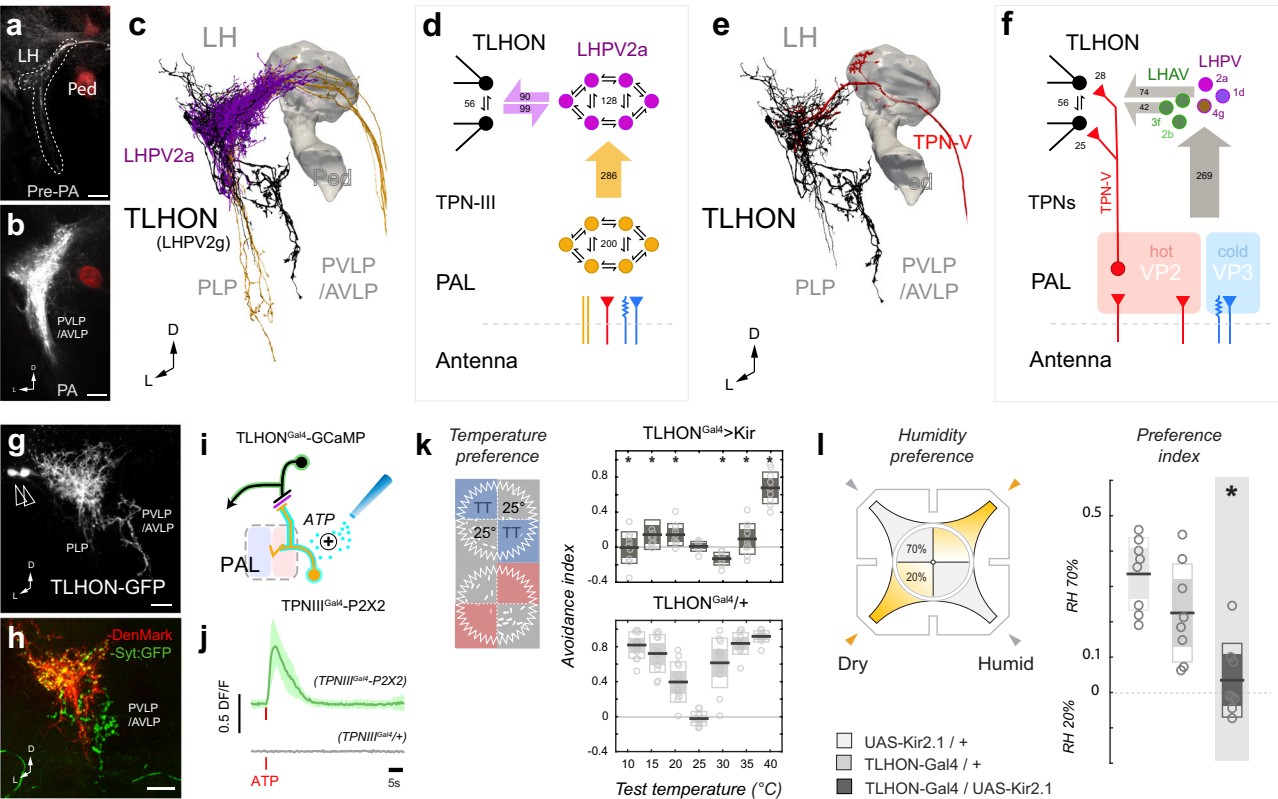

**Fig. 2 | TLHONs are key TPN-III functional targets in the Lateral Horn. a,b** Photo-activation of PA-GFP illuminates postsynaptic target regions of TPN-III. **a** Pre-PA 2-photon stack, TPN-III axons are stably labeled with CD8:GFP, the MB is labeled by independent expression of DsRed (Ped, red) as a landmark. **b** Post-PA 2-photon stack; labeled cells densely innervate the posterior ventral lateral horn; projections to the PVLP/AVLP are also visible (arrowhead); images represent 5 independent repeats. **c,d** Connectomic analysis reveals TPN-III directly target LHPV2a inter-neurons which are inter-connected with TLHONs and, in turn, display prominent projections to the PVLP/AVLP. **c** 3D reconstruction of five TPN-III axons (orange), the six reconstructed LHPV2as and the two TLHONs (the MB Peduncle is shown as a landmark). **d** Connectivity diagram from TPN-IIIs (orange) to LHPV2as (purple) to TLHONs (black). Large arrows represent synapses between cell types, small arrows represent connections between cells of the same type (numbers = synapse counts). **e,f** TLHONs have additional direct and indirect thermosensory drive. **e** 3D reconstructions of TPN-V (red) providing direct thermosensory input into TLHON (black). **f** Connectivity diagram. TPN-V is a TPN with dendrites in the hot glomerulus (also known as VP2) of the PAL. Additional indirect thermosensory drive may derive from LHAV/LHPV cell types that are targeted by TPNs and in turn target TLHONs (numbers = synapse counts, circles and letters = cell types, as in 2a = LHPV2a etc.) **g,h** A selective driver for TLHON. **g** TLHON-Gal4/UAS GFP shows prominent GFP expression in the two TLHONs (arrowheads = cell bodies). **h** Co-expression of a dendritic (red) and synaptic (green) marker confirms TLHON polarity; images represent 7 (**g**) or 3 (**h**) independent repeats. **i,j** TLHONs are functional targets of TPN-IIIs. **i** Experiment schematic. **j** Calcium responses recorded by GCaMP imaging in TLHONs after artificial activation of TPN-IIIs by the ectopic expression of ATP receptor P2X2 ($N = 5$ cells; lines and shading = mean ± SD). **k** Silencing of TLHONs affects hot and cold avoidance behavior in 2-choice tests (black/red line = mean, inner box = 95% CI, outer box = SD; individual points = # of groups; $N = [10,12,16]$ groups at each temperature for Gal4/Kir, Ga4/+, Kir/+, respectively, 20 flies per group; * = p < 0.05, p = [5.82E−17, 2.18E−09, 2.13E−04, 0.227,1.81E−13,1.18E−16,2.80E −07], for 25 °C vs 10,15,20,25,35,40 °C, 2-way ANOVA). **l** TLHON silencing also affects dry air avoidance in a four-field test for humidity preference (* = p < 0.05, p = 1.57E−04, 2-way ANOVA); **k** and **l**: black line = mean, inner box = 95% CI, outer box = SD; individual points = # of groups, $N = 8$ groups, 20 flies per group for each genotype. PLP=posterior lateral protocerebrum, Ped= mushroom body peduncle, LH=lateral horn, LHPV = lateral horn posterior ventral, PVLP = posterior ventral lateral protocerebrum, AVLP = anterior ventral lateral protocerebrum, PAL = posterior antennal lobe, TPN = temperature projection neuron, VP = ventral posterior antennal lobe, TT = test temperature. **a,b,g,h**: scale bars = 10 μm.) Source data are provided as a Source Data file.

behavioral responses to a battery of odors in a 4-field olfactory assay (including odors that activate VL2p, VM4 or VL2a). Importantly, even in this same 4-field arena (and in the presence of air flow) TLHON silencing robustly impacted hot and cold avoidance (Fig. 3c–e), while the avoidance of a number of innately aversive odors was not affected (Fig. 3f, g, and see controls in Fig. 3h), suggesting TLHONs are not broadly required for aversive navigational responses.

Together, our results suggest that TLHONs are an important functional output for TPN-IIIs, and that silencing their activity has selective effects on the avoidance of hot and cold temperature (as well as on the avoidance of dry air).

### TLHON firing tracks the rate of temperature change
Our experiments so far demonstrated functional connectivity between TPN-III and TLHONs, and our silencing results showed a similar, broad impact on temperature preference behavior resulting from the silencing of each cell type. Connectomics also highlighted the poten-tial for a broader drive from thermosensory circuits onto TLHONs.

Our next goal was to characterize the response profile of TLHONs to a battery of thermal stimuli, again using 2-photon guided patch clamp electrophysiology and step stimuli in the hot and cold range. Interestingly, the firing pattern of TLHONs in response to thermal stimulation was similar but not identical to that of TPN-IIIs.

Similar to TPN-IIIs, TLHONs responded to heating or cooling with a burst of action potentials (ON response, Fig. 4a, b, arrowheads in b) that was independent of absolute temperature (Fig. 4c). Unlike TPN-IIIs though, TLHONs' steady-state firing appeared modulated by absolute temperature, so that firing was generally higher in cold temperature (<25 °C, Fig. 4c and see quantification in d).

Next, we looked in more detail at the temporal dynamics of TLHON activation. Interestingly, during our standard heating or cool-ing step stimulus (-5 °C Δt from 25 °C), the peak TLHON membrane

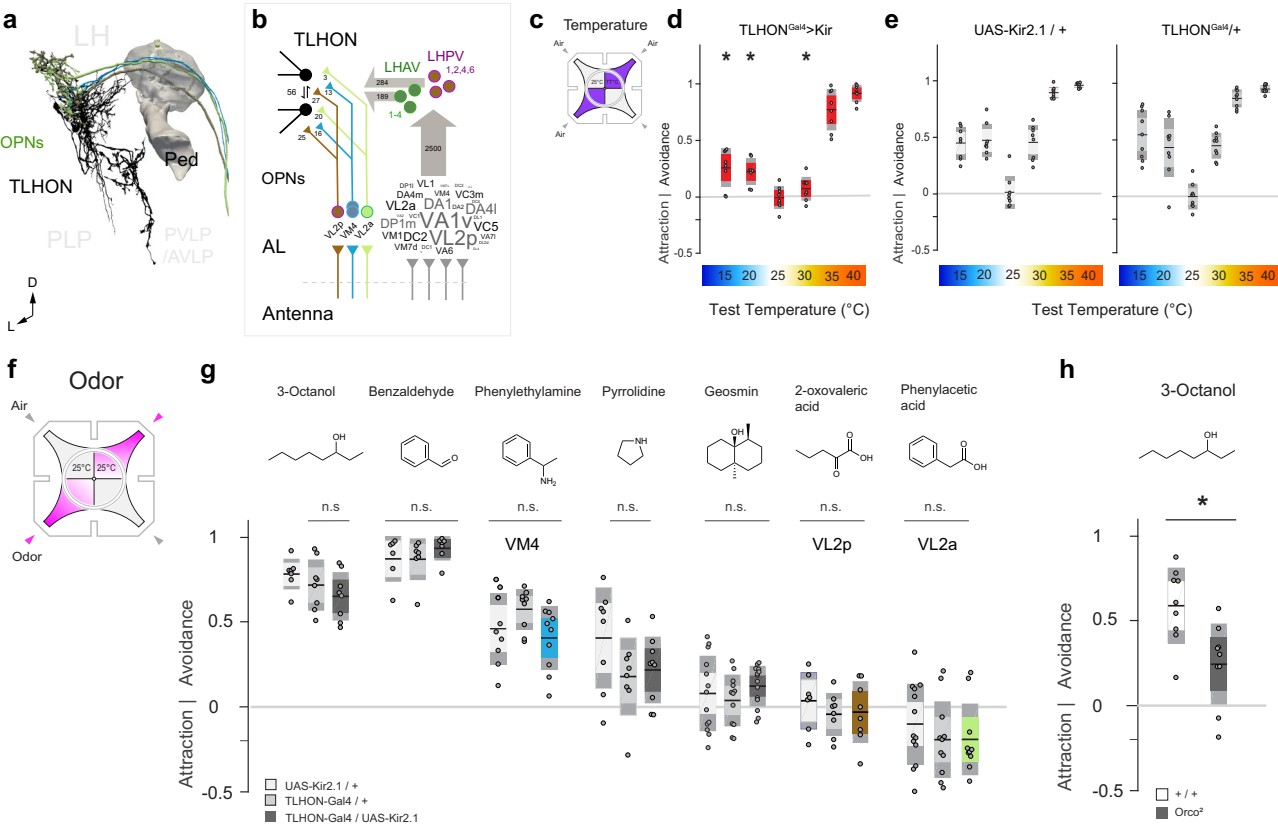

**Fig. 3 | TLHON activity is required for thermal preference but not for olfactory responses. a,b** Connectomic analysis demonstrates potential olfactory drive on TLHONs from four uniglomerular OPNs. **a** 3D reconstruction of three TPN-III axons (green, cyan, and brown) that provide direct input onto TLHONs (black; part of the MB is shown as a landmark). **b** Diagram of connections from uniglomerular OPNs to TLHONs. Left: direct connections, OPNs are labeled by the glomerulus they innervate; Right: connections involving a single intermediate cell type. OPNs representing ~30 glomeruli innervate intermediate LH neurons in the LHAV/PVs (for OPNs, the relative synaptic contribution of each glomerulus corresponds to the size of the label as calculated by a word cloud function, range = 10-393 synapses; gray arrows = aggregate # of synapses, circles and letters = cell types, as in 1 = LHPV1, etc.). **c-h** A four-field 2-choice assay designed to test temperature and olfactory preference in the same environment in the presence of air flow (gray/purple). **c** Thermal preference assay schematic. **d** Silencing TLHONs results in hot and cold avoidance deficits. **e** Genetic controls (N = [8,9,8] groups for each

temperature for Gal4/Kir, Ga4/+, Kir/+, respectively, 20 flies/group; * = p < 0.05, p = [0.012, 0.028, 0.934, 2.22E−06, 0.106,0.206] for 25°C vs 10,15,20,25,35,40°C, respectively, 2-way ANOVA). **f** Olfactory preference schematic (gray= air plumes, pink = odor plumes). **g** Genetic silencing of TLHONs has no impact on flies' responses to the odors tested, including odors that activate glomeruli innervated by direct OPNs (green, cyan, and brown as in **b**; 20 flies/group, N =[8,8,7], [7,7,6], [10,9,9], [9,8,8], [14,12,13], [8,8,7], [10,11,14] groups for Gal4/Kir, Ga4/+, Kir/+, respectively, corresponding to odors: 3-Octanol, Benzaldehyde, Phenylethylamine, Pyrrolidine, Geosmin, 2-oxovaleric acid, Phenylacetic acid, respectively; p = 0.006, 2-way ANOVA, n.s. = not significant. **h** As a control, Orco2 mutants have a strong deficit in olfactory avoidance compared to controls (N = 9 groups, 20 flies/group; * = p < 0.05, p = 0.006, 2-sample, 2-tailed t-test). For all boxplots **d–e**,**g–h**: black line = mean, inner box = 95% CI, outer box = SD; individual points = # of groups). Source data are provided as a source data file.

---

potential (corresponding to the peak of TLHON's "ON" response) preceded the maximum or minimum absolute temperature reached, and did not correlate with the maximum thermal change (Δt) experienced by the preparation. Instead, TLHON maximum depolarization correlated with the peak of the first derivative of the stimulus, i.e., the fastest rate of thermal change in either direction (heating or cooling; Fig. 4e).

Interestingly, TPN-III responses also correlated well with the fastest rate of thermal change (the first derivative of the stimulus, Supplementary Fig. 2). Indeed, when exposed to a battery of thermal stimuli from different baselines and characterized by different rates of thermal change, both the firing rate of TLHONs and that of the upstream TPN-IIIs (at the peak of the ON response) broadly correlated with the rate of thermal change in both heating and cooling directions (Fig. 4f−h). Yet the response profile of the two cell types was again not identical (compare Fig. 4g and h): while TPN-IIIs' firing rate scaled rather smoothly with the rate of temperature change (Fig. 4h), TLHON responses appeared to have a threshold, so that TLHON firing was elicited by rapid (>0.2 °C/second) but not slow thermal change (Fig. 4g

and see Fig. 4i,j for within-cell, baseline-subtracted responses to slow and fast thermal change and Fig. 4k for representative TLHON traces).

### TLHONs responses prime fly behavior for heat escape

What may be the significance of this "derivative" signal to heat avoidance behavior? We previously characterized rapid heat avoidance in freely moving single flies with high spatial and temporal resolution[28]. This work made use of a realistic 3D simulation of the thermal environment to show that, in rapid 2-choice assays, flies encounter very steep thermal gradients at the boundaries between the home (~25 °C) and test quadrants (set at 30°, 35°, or 40 °C[28], and Fig. 5).

Whenever the fly encounters the cool/hot boundary, temperature preference behavior manifests as the probability that the fly will cross over into the hot quadrant or rather produce a sharp turn away from it. We demonstrated that the probability of an escape turn is a function of the test temperature (the higher the TT the more likely a fly is to turn away) and that the flies use differential temperature reading from the two antennae to chart a suitable trajectory for escape (i.e., they are more likely to escape by turning left if the right antenna is exposed to more intense heat, and vice versa). Importantly, we also showed that,

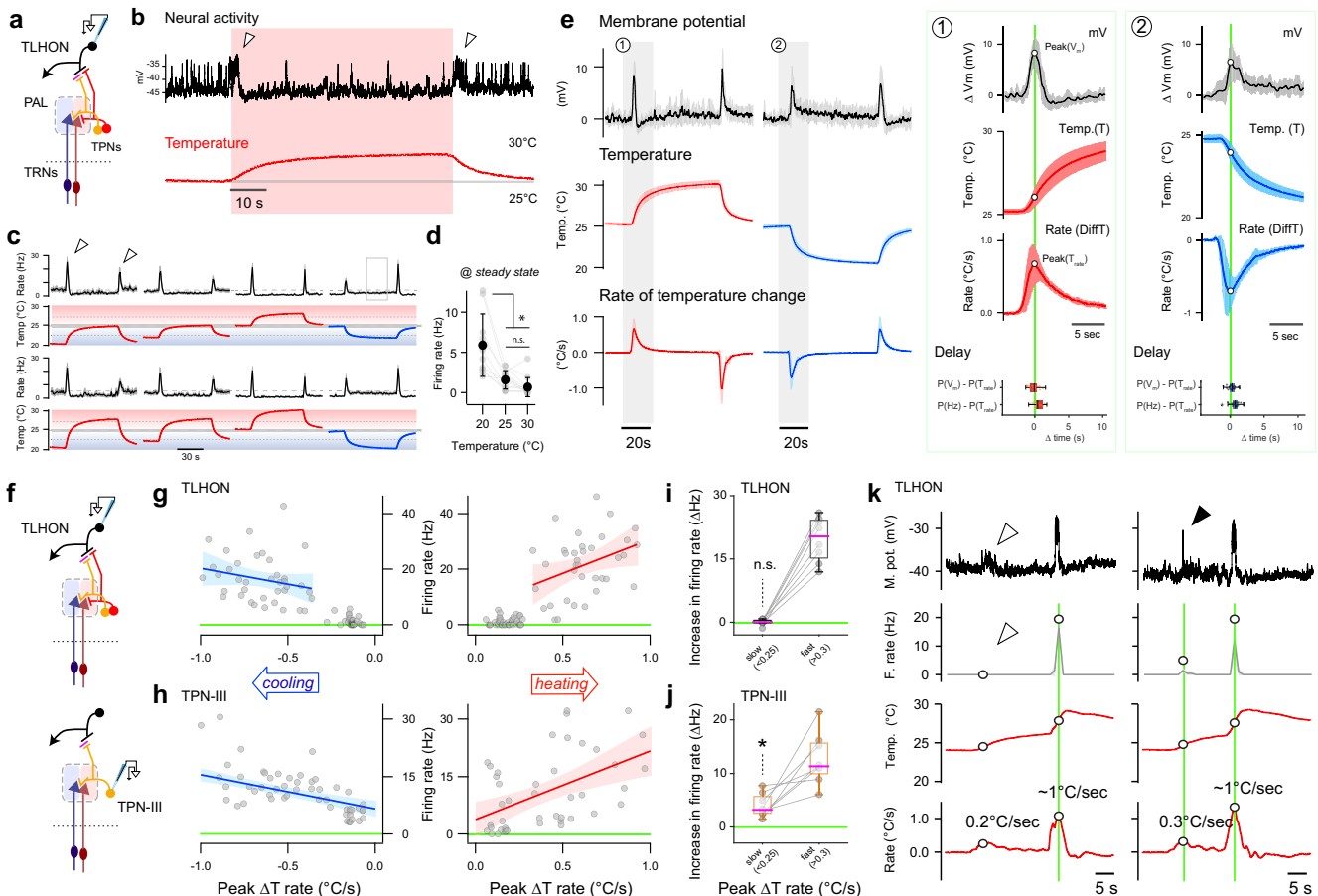

**Fig. 4 | TLHON ON responses report fast heating or cooling. a–d** 2-photon guided patch clamp electrophysiology of TLHONs reveals robust ON responses to heating and cooling and some modulation by absolute temperature. **a** Recording schematic. **b** Representative whole-cell current clamp recording (top; arrowheads = ON responses) and corresponding thermal stimulus (bottom, red). **c** Average firing rate histograms of TLHONs in response to a battery of temperature stimuli. ON responses (arrowheads) are independent of absolute temperature. **d** Steady-state firing rate quantifications demonstrates modulation by absolute temperature (**c** and **d**: $N = 10, 13, 13$ cells, 10 animals, line and shading in (**c**) indicate mean ± SEM, gray circles connected by lines in (**d**) indicate recordings from the same cells; black circles indicate mean ± SD; * = p < 0.05, p = 9.81E−06 1-way ANOVA, n.s. = not significant). **e** TLHON ON responses peak in correspondence with the fastest rate of heating or cooling. TLHON filtered membrane potential responses (top traces), stimulus temperature (middle traces), and rate of thermal change (temporal derivative of temperature stimuli; bottom traces). Green boxes to the right are x-axis expansions of gray shaded regions as indicated. Boxplots show temporal locking between peak response and peak stimulus rate. Delay (shown as boxplots) is quantified as the difference between the timing of the peak membrane potential or the peak firing rate and the timing of the peak rate of thermal change ($N = 11$ cells/

2 samples per cell/8 animals, trace line and shading indicate mean ± SEM; boxplots: black line = median, box = interquartile range, whiskers = range, dots = individual cells). **f–k** TPN-III ON responses track the rate of temperature change over a broad range, while TLHONs respond to fast thermal change with an apparent threshold of >~0.2 °C/sec. **f** Schematics of recordings. **g,h** Peak firing rates for TLHONs (**g**) and TPN-IIIs (**h**) plotted against the maximum rate of temperature change of the corresponding stimulus ($r^2$ from left to right (**g**) 0.05, 0.13; (**h**) 0.35, 0.25; shading: ± 95% CI; gray circles: mean firing rate from a single cell during a single stimulus sweep, see methods for details; $N = 176$ data points from 20 cells/12 animals for TLHONs and $N = 115$ data points from 16 cells/8 animals for TPN-III). **i,j** Sequential responses from (**i**) TLHONs or (**j**) TPN-IIIs challenged with slow followed by fast thermal change (8 cells/6 animals for each, 4 heating and 4 cooling stimuli, responses from the same cells are connected; firing rates are background-subtracted; *= different from zero p = 0.001, n.s.= not different from zero p = 0.873; 1 sample, 2-tailed *t*-test; boxplots: pink line = median, box = interquartile range, whiskers = range, dots = individual cells). **k** Representative TLHON membrane potential traces (top, black) and firing rate histograms (gray) in response to sub-threshold (empty arrowhead) and suprathreshold (filled arrowhead) stimuli. Source data are provided as a source data file.

during each experiment, flies quickly learn to turn earlier within the thermal gradient to minimize heat exposure, particularly when the experiment involves "dangerous", hot test temperatures (35°–40 °C)[28].

Here, our explicit hypothesis was that a "derivative" signal such as the one encoded in the activity of TLHONs may be flexibly used to modulate the response to heating depending on context, i.e., to trigger an early escape turn if stimulus trajectory and/or learning[28] suggest that dangerous conditions are imminent.

To test this idea, we again silenced the activity of TLHONs by transgenic expression of the hyperpolarizing agent Kir2.1, and recorded navigational responses from single flies at high spatial and temporal resolution as described before[28] (note that we also used this high-resolution assay to run a number of additional controls to evaluate

potential confounding effects due to off-target expression of TLHON-Gal4, see supplementary Fig. 3 and legend).

As expected, genetic silencing of TLHONs severely impaired heat avoidance behavior in this assay. Similar to the group assays, TLHON silencing abolished single-fly avoidance of 30 °C and reduced avoidance of 35 °C (Supplementary Fig. 3). Fly trajectories revealed frequent cool/hot boundary crosses and a sharp increase in the likelihood a fly would enter hot quadrants (Fig. 5a–c).

This phenotype is reminiscent of that of flies with impaired heat detection[28] but, importantly, not identical: flies in which the hot receptors of the arista have been silenced (or the whole antenna has been surgically removed) cannot produce the appropriate ballistic left/right escape trajectories, but instead turn with equal probability to

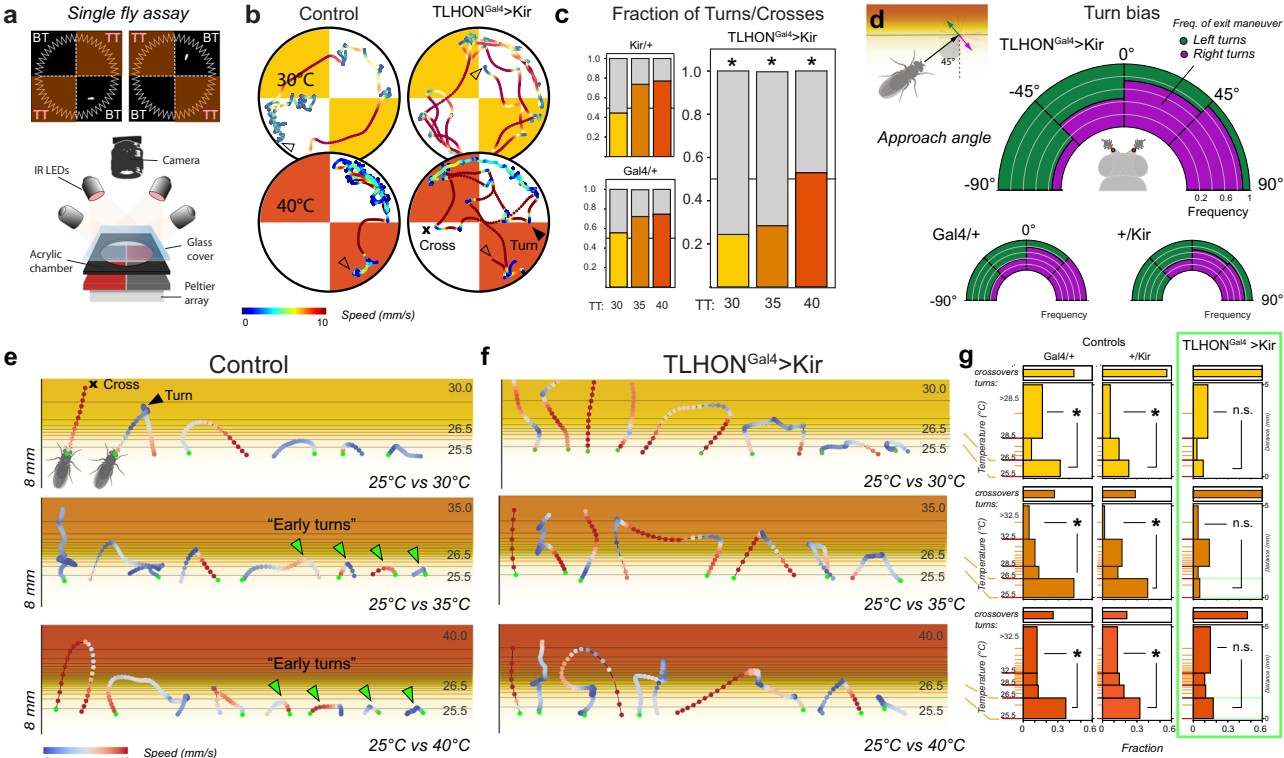

**Fig. 5 | TLHON responses prime fly locomotor behavior for rapid heat escape ("early turns"). a** Schematic of the single-fly 2-choice assay for temperature preference. BT = base temperature TT= test temperature. **b,c** TLHON silencing affects heat avoidance. Heat avoidance is expressed by turning away from heat at the cool/hot border (black arrowhead), rather than crossing into the hot quadrant (black cross). **b** Representative trajectories of control flies and of TLHON-silenced flies in 2 different experimental conditions: 25° vs 30 °C (mild heat, yellow) and 25° vs 40 °C (noxious heat, orange; trace color: fly speed). **c** Fraction of U-turns to border crosses at the cool/hot boundaries. TLHON-silenced flies are less likely to produce escape turns at the cool/hot border than controls in the 25° vs 30 °C, 25° vs 35 °C and 25° vs 40 °C conditions (* = p < 0.05, p = [0.001, 5.6E−07, 3.28E−05] for 25 °C vs 30,35,40 °C, respectively; GLMM, two-sided Wald Test Wald Test; N30 = [32,22,35], N35 = [27,25,26], N40 = [27,36,27] flies for Gal4/Kir, Ga4/+, Kir/+, respectively). **d** TLHON silencing does not affect turning bias at the border. Flies use differential temperature readings from the antennae to produce escape turns so that the angle of heading is predictive of the angle of escape. In (**d**) the heading angle is quantified in relation to the isothermal lines of the cool/hot boundary while the escape turn is categorized as "left" (green) or "right" (purple) and binned in 45° intervals. Fan plots represent the distribution of left/right escape turns as a function of initial heading angle and are not significantly different between control and experimental groups (GLMM, two-sided Wald Test, p = 0.674, N = [27,36,27] for Gal4/Kir, Ga4/+, Kir/+, respectively; data from 25 °C vs 40 °C experiments). **e–g** Normal activity in TLHONs is required to produce "early turns" in response to exposure to hot conditions (>35 °C). **e,f** Representative trajectories of border interaction events for control (**e**) and TLHON-silenced flies (**f**). Trajectories are colorized based on fly speed and overlaid on the thermal gradients predicted to form at the boundary between the 25° vs 30 °C, 25° vs 35 °C and 25° vs 40 °C floor tiles (see text). TLHON-silenced flies produce less "early turns" in the 25° vs 35 °C and 25° vs 40 °C conditions (green arrowheads) and instead turn deeper into the gradient. **g** Quantification of turning frequency within distinct bands of the thermal gradients. "Early turns" ( < 26.5 °C) and "deep turns" (>28.5 °C or >32.5 °C) are compared within genotype to quantify the propensity for rapid escape (* = p < 0.05, p = [0.042,1.95E−05, 3.5E−4], p = [7E-5, 9.84E−10,0.001], p = [0.336,0.593,0.631], for Gal4/+, +/Kir, Gal4/Kir, and 25 °C vs 30,35,40 °C, respectively, n.s. = not significant, p > 0.05; two-sided Chi square Test, N = 22–36 flies; see statistics table for precise N values for each test). Source data are provided as a source data file.

the left or right irrespective of their angle of approach to the heat gradient[28].

In contrast, while TLHON-silenced flies perform fewer escape turns at the boundaries, the left/right distribution of such turns appears normal, in that it is determined by the angle of approach as is that of control flies (Fig. 5d). This suggests that their heat sensing, steering, and navigation abilities are not impaired.

Instead, analysis of border trajectories demonstrates that TLHON-Kir flies react less readily when they encounter the thermal gradient. As mentioned above, within each experiment wild type[28] and control flies learn to react quickly when encountering the cool/hot boundary (particularly in the 25 vs 35 °C and 25 vs 40 °C conditions[28]), and therefore often turn back as soon as they experience heating ("early turns", Fig. 5e–g). In contrast, TLHON-silenced flies wander deeper into the gradient and spend more time exposed to heat even in cases in which they eventually perform an escape turn (Fig. 5f, g).

We interpret this result as supporting the idea that the TLHON "ON" response may represent an early alarm signal that alerts the

fly that a rapid temperature change is going to bring about dangerous thermal conditions, priming locomotor behavior for an escape maneuver (an early turn). In the absence of this alarm signal (when TLHONs are silenced), flies are still able to perform escape turns, but they do so at lower frequency and often deeper into the thermal gradient.

**Modeling the impact of a derivative signal on fly behavior**
Our next goal was to model the potential impact of a circuit performing a temporal derivative transformation similar to the one recorded in TLHONs on fly heat avoidance behavior. We previously created an in silico vehicle-model scaled to the physical dimensions of the fly. This agent was inspired by a classic Braitenberg vehicle and we used it in virtual 2-choice assays to probe the notion that simple heat avoidance may be understood as a combination of hard-wired responses[28]. In fact, comparison of the performance of this vehicle to that of real flies was instrumental in revealing behavioral plasticity and the learning effects described above.

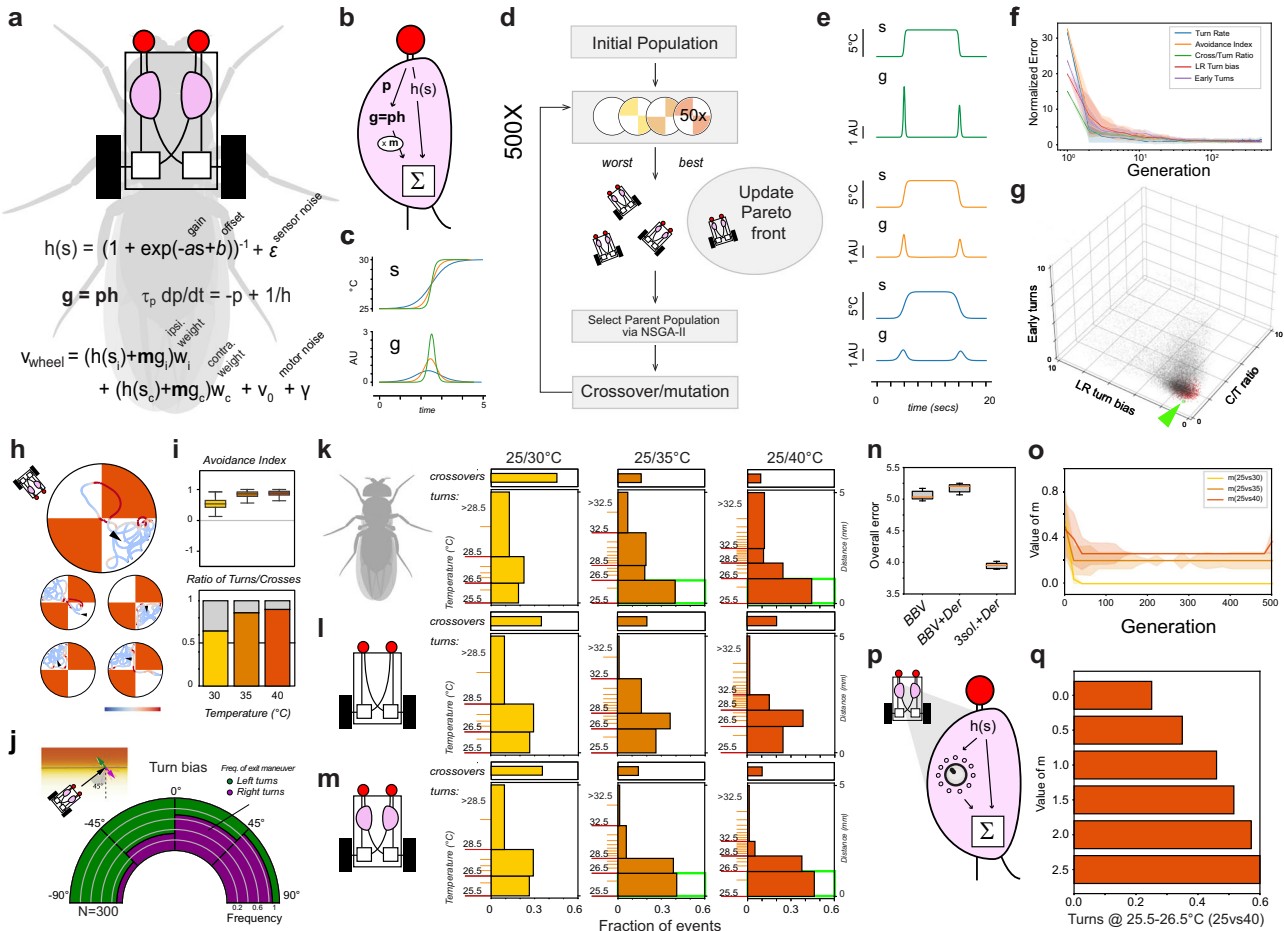

**Fig. 6 | A simple vehicle model incorporating a tunable "ON" signal reproduces flexible heat avoidance behavior. a-d** An in silico Braitenberg vehicle model based on the dimensions of the fly, incorporating a simple "brain": a flexible central processing step designed to mimic an ON response tracking stimulus rate, acting in parallel to the relay of signal from the sensor to the motor (red). **a** Basic model design, with key parameters used as substrate for evolution (s = sensory input, v = wheel velocity; parameters: $a$ = gain, $b$ = offset, $\varepsilon, \gamma$ = noise, $w_j, w_c$ = weights of ipsi- and contralateral connections, respectively, $m$ = multiplier, $\tau_p$ = time constant). **b** Brain design. **c** Conceptualization of the transformation performed by the filter for signals of different rate (s = input signal, g = output) **(d)** Schematic of the evolutionary process used to optimize the parameters. Only the multiplier, $m$, is allowed to vary between experimental conditions (25° vs 30 °C, 25° vs 35 °C and 25° vs 40 °C). **e** Input-output transformation at the central processing step following evolution, applied to representative signals of different rate (s = input, g = output). **f** Error convergence for the five different objectives as a result of evolution (median ± median absolute deviation; error values are from each generation's Pareto front vehicles; the error of each vehicle is normalized by the median error of the final Pareto front vehicles). **g** 3D scatter plot showing the error space for 3 key objectives for all vehicles tested (gray), the all-time best performing vehicles after 500 generations (red), and a chosen top performer (green, chosen at random from a group of top performers, see methods for details). X-axis = Crossover/U-turn ratio error,

Y-axis = early turn frequency error, Z-axis = Left/Right turn predictability error. **h–j** The evolved top performing vehicle (green dot in **g**) nearly reproduces fly thermotactic behavior in a simulated arena. **h** Vehicle trajectories in the simulated arena (25° vs 40 °C, arrowhead = start). **i,j** Vehicle performance parameters (n = 400 simulations), **i:** boxplots: black line = median, box = interquartile range, whiskers = range. **k–n** A flexible "brain" is essential for the vehicle to reproduce the appearance of early turns in the 25° vs 35 °C and 25° vs 40 °C conditions. **k–m** Quantification of turning frequency within distinct bands of the thermal gradients in (**k**) real flies (N = [25,24,25] for 25° vs 30°,35°,40 °C, respectively), (**l**) a top performing evolved Braitenberg vehicle and (**m**) a top performing evolved vehicle in which the parameter, m, is allowed to reach 3 different solutions for the 3 experimental conditions. Only the latter vehicle displays appropriate early turns, similar to the real fly, and (**n**) can minimize early turn error; boxplots: orange line = median, box = interquartile range, whiskers = range (n = 1056 top performers). **o** Across evolutionary time the influence of the central filter (parameter m) settles on values proportional to the frequency of early turns in each experimental condition (n = 112 vehicles/generation, median ± median absolute deviation). **p, q** Independently increasing the value of m (i.e., the influence of the central processing step) increases the frequency of early turns (n = 200 simulations each). Source data are provided as a source data file.

In our first iteration, the vehicle-model consisted of two antennal sensors connected by two wires each to both an ipsilateral and contralateral motor[28]. The model included eight free parameters and we used an evolutionary algorithm to select a set of solutions that could best approximate performance metrics of heat avoidance behavior of real flies in a set of 25° vs 30 °C, 25° vs 35 °C and 25° vs 40 °C tests.

This simple model was intentionally "hard-wired" (a single solution was tested across all experimental conditions) and lacked a "brain", in that the only transformation of the stimulus was actuated by a simple equation in the antennal sensor[28].

Our approach here was to add an additional "central" processing step (Fig. 6a–c), to model this filter on the "ON" activity of TLHONs (Fig. 6c), and to flexibly combine its output in a simple brain in parallel to the signal propagating from sensors to motors (Fig. 6b). Our goal was to explicitly test the possibility that varying the influence of this "ON" filter may produce flexible behavior –namely reproduce the occurrence of early turns observed in wild type flies as a function of test temperature.

As before, we used an evolutionary algorithm to optimize model parameters (ref. 28, Fig. 6d–j) but this time we allowed the system to

reach 3 solutions (one for each experimental condition: 25° vs 30 °C, vs 35 °C and vs 40 °C), differing in a specific multiplier representing the influence of the "derivative" filter.

Our results suggest that introducing this simple and tunable central processing step produces vehicles that can both approximate fly heat avoidance behavior and reproduce the differential occurrence of early turns as a function of test temperature observed in wild-type flies (Fig. 6k–q).

Parameter tuning was indeed essential to accomplish flexible behavior. First, our simulations demonstrate that no single solution could effectively match the pattern of early turns observed in real flies (Fig. 6k, l, n). Second, during the course of 500 generations of simulated evolution, the multiplier that represents the influence of the derivative ON signal settled on values that were proportional to the occurrence of early turns (Fig. 6o) and, third, an increase in early turns could be independently accomplished by increasing the influence of the filter (increasing the multiplier; Fig. 6p, q).

Together, our results support the notion that an ON "derivative" signal such as the one we observe in TLHONs can be flexibly used to modulate the response to temperature change, such that a small heating step may become salient, and induce a significant escape response (an early turn) if external conditions (imminent thermal danger) or previous experience (learning) suggest to the animal this may be an adaptive strategy.

### TLHONs display privileged connectivity with descending pathways that mediate escape

Our ultimate goal is to understand how the central processing of thermal stimuli influences locomotion to produce directed responses such as thermotaxis and heat avoidance. The availability of the fly connectome provided us with an opportunity to begin to study how TLHON responses may propagate to pre-motor and motor circuits, to ultimately prime behavior for escape turn initiation and avoidance.

The top-down control of fly locomotion relies on a population of -350–500 descending neurons (DNs), with cell bodies in each hemisphere of the brain and axons that target the ventral nerve cord (VNC[29,30]). Our exploration of the hemibrain connectome did not identify any DN immediately post-synaptic to TLHONs (Fig. 7a, b; threshold: 10 synapses). Instead, analysis of the top 10 synaptic outputs of TLHONs revealed a dense indirect pattern of connectivity to DNs mediated by just two Lateral Horn (LHAD1, LHAV2b) and two Ventro-Lateral Protocerebrum neurons (PVLP076, AVLP053; Fig. 7b,c and see Supplementary Table 3 for IDs). We identified a total of 11 DN cell types that could receive indirect input from TLHONs through these intermediate neurons. Ten of these are in a DN cluster residing on the posterior surface of the brain (DNp01-06, DNp09, DNp11, DNp13, DNp32), and one is in the Anterior Ventral cluster (DNb05).

Strikingly, the majority of these DNs have been reported to target leg-associated neuropils as well the lower and intermediate tectulum, regions of the VNC that are thought to coordinate leg movements during locomotion (Fig. 7a, b; ref. 30). Previous work has implicated a number of these DNs in jumping/escape behavior (DNp01[31]; DNp02,04,11[32]), in fast locomotion (DNp01,02,05,11,13, DNb05[33]), and defensive responses (DNp06[34], DNp09[35]). In contrast, DNs that have been associated with slow locomotion (DNa05,07, DNb02, DNd02,03; DNg03,15[33]), grooming (DNg07,08[33]; 11,12[36]), or abdomen/wing movement (DNg02,11,30, DNp10, DNp27,29, pIP10[33]) were not found as downstream targets of TLHONs using these criteria.

In fact, the most prominent indirect downstream targets of the TLHONs (Fig. 7b, thick lines and circles) have been directly implicated in escape behavior and in the production of evasive turns. DNp01 (Fig. 7c), also known as the Giant Fiber neuron (GF), innervates the lower tectulum and is well-studied for its role in rapid escape jumps in response to looming visual stimuli, mechanical and olfactory startle[31].

DNp06 (Fig. 7c) appears as a prominent downstream target of TLHONs, as it is directly downstream of both LHAD1 (336 synapses) and PVLP076 (145 synapses; Fig. 7b, c). Interestingly, normal activity in DNp06 has been recently shown to be important for evasive flight maneuvers in response to approaching visual objects[34].

While an involvement of DNp01/giant fiber in our 2-choice behavior appears unlikely (as GF activation is usually associated with jumping), DNp06 has been previously involved in escape turns[34]. Therefore, we next tested the possibility that genetic silencing of DNp06 may produce navigational phenotypes consistent with a role downstream of the TLHONs during heat avoidance.

Our results suggest that genetic silencing of DNp06 produces a specific navigational phenotype similar to what we described for TLHONs, in which flies consistently fail to produce "early turns" when encountering the thermal gradient and instead turn deeper, in correspondence to higher temperatures (Fig. 7d). Silencing of an additional (more weakly connected) downstream DNp, DNp09, produced no such defects (Fig. 7e), suggesting DNp06 (perhaps together with other DNps) may indeed play a key role downstream of TLHONs during heat avoidance behavior.

On the whole, both the privileged (albeit indirect) connectivity of TLHONs with DNps and the results of DNp06 genetic silencing are consistent with the notion that DNps may contribute to propagate TLHON activity to centers of the VNC that control fast locomotion, turning, and escape, helping to prime behavior for heat avoidance maneuvers when dangerous conditions are imminent.

## Discussion

Here, we identify a full circuit—from second-order thermosensory neurons, to output neurons of the lateral horn (TLHONs), to descending cells that target the ventral nerve cord—that appears to function as an early warning system during navigation of steep thermal gradients (Fig. 7f). ON responses in this circuit alert a moving fly that potentially dangerous heat may be looming, and allow it to prepare for a rapid response.

Rapid temperature change is surely not always dangerous for a fruit fly (e.g., rapid heating will likely have a positive connotation in the cold). Yet the fly is a tiny poikilotherm with a very small thermal capacity, so that rapid heating or cooling can easily mean comfortable temperatures can quickly turn into dangerous conditions that can incapacitate or even kill the fly within minutes if not seconds.

For example, considering a rate of thermal change of 0.5°C/s (just above TLHON's threshold for ON responses, see Fig. 4), a stationary fly at 25 °C would experience -10 °C of thermal change within 20 s, quickly risking exposure to noxious (>35 °C) or even potentially lethal temperatures (>40 °C). A moving fly can encounter even faster rates of temperature change: in our arenas flies walk at -5–10 mm/second and can experience thermal change in excess of -5–10 °C/second or more[28] (flying flies move even faster, and can cover a remarkable -80 cm/second[37]).

Rapid thermal change is therefore likely a very salient stimulus for a fly in most situations. Our hypothesis is that in the context of heat avoidance "salient" overlaps with dangerous and promotes "turn away from" behavior. In different conditions (e.g., heating from a cold baseline) rapid temperature change may signal favorable conditions and promote "turn toward" behavior.

Consistent with this idea, TLHON downstream circuits connect broadly with a set of DNs that influence a range of responses including fast locomotion, turning and escape, rather than forming exclusive connections with neurons that control ipsi/contra-lateral turning behavior (as in a simple escape reflex circuit). We believe this organization may reflect flexibility on the behavioral output associated with the salient "rapid thermal change" stimulus.

ON responses are widespread across sensory systems, both at the receptor level and in central pathways that process sensory

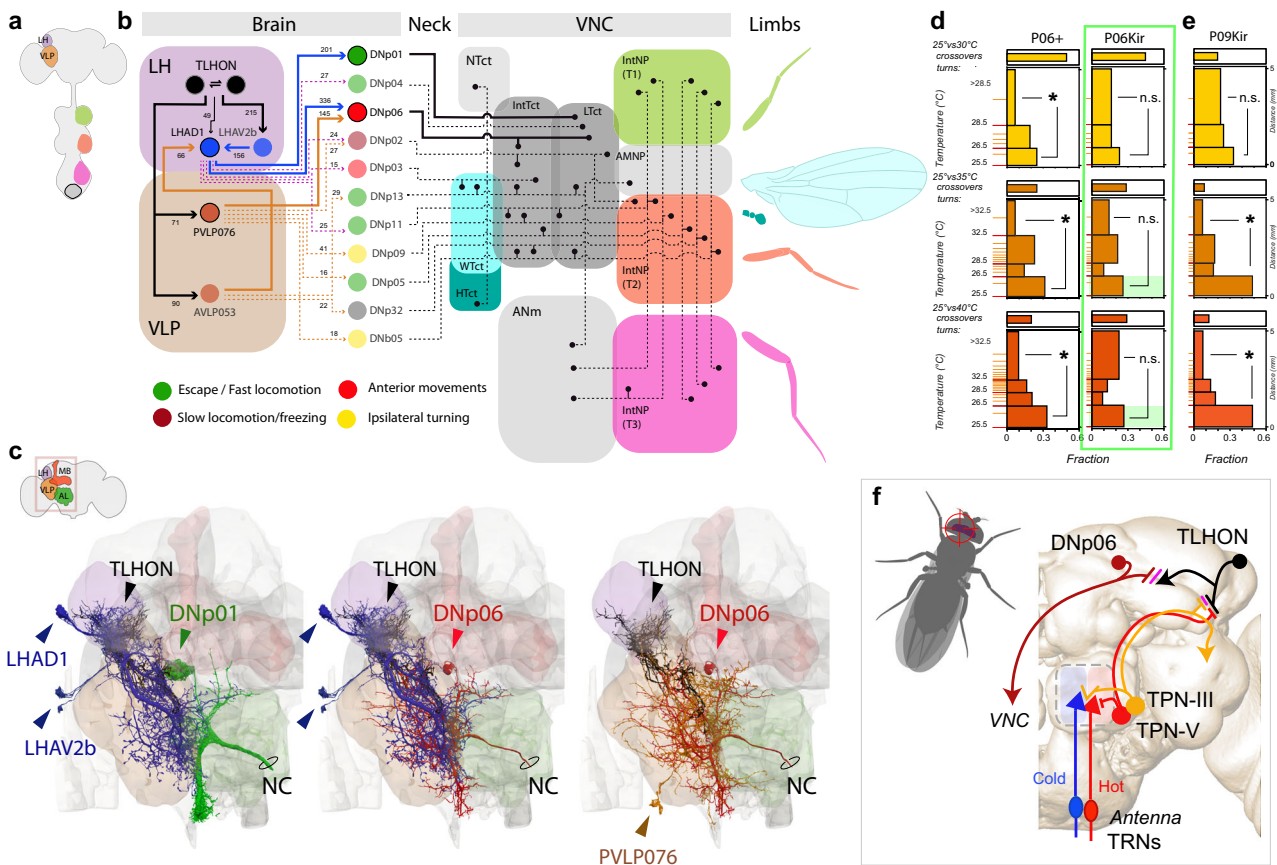

**Fig. 7 | TLHONs display privileged connectivity with descending pathways that mediate escape. a,b** Connectomic analysis centered on potential descending pathways downstream of TLHONs based on Hemibrain data. No direct TLHON > DN connections were found. Additional pathways including a single intermediate cell type were considered (threshold: 10 synapses). Cells belonging to the same cell type were treated as a unit. **a** Brain-VNC schematic, significant neuropils are color coded as in (**b**–**d**), leg and wing neuropils are color coded as the corresponding associated body part. Arrows = synaptic input, numbers = synapse #. **b** In the brain, TLHONs connect to two interneurons (LHAD1 and PVLP076) with privileged connectivity with DNp-type descending neurons. LHAD1 also receives additional recurring connections from TLHON through LHAV2b and AVLP053. Together, LHAD1, PVLP076 and AVLP053 connect to 10 DNp cell types and one DNb. The strongest connectivity is observed with DNp01 (201 synapses) and with DNp06 (336 synapses from LHAD1 and 145 from PVLP076). In the VNC, the identified DNps connect preferentially with neuropils associated with locomotion, turning and escape. Neck/VNC black lines represent DN targets based on literature (see text for details). **c** EM reconstructions of TLHON > LH/VLP>key DNps (LH, VLP and cell

types are color coded as in (**a**); neuropil colors are schematized in (**a** and **c**) top left, arrowheads = cell bodies). **d** Silencing DNp06 reduces the occurrence of "early turns" in the 25° vs 35 °C and 25° vs 40 °C conditions (green box), while (**e**) silencing DNp09 does not. Sideways histograms represent quantification of turning frequency within distinct bands of the thermal gradients in the 25° vs 30 °C, 25° vs 35 °C and 25° vs 40 °C conditions, for controls (Gal4/+, left) and experimental animals (Gal4/UAS-Kir2.1, right). "Early turns" (<26.5 °C) and "deep turns" (>28.5 °C or >32.5 °C) are compared within genotype to quantify the propensity for rapid escape (two-sided Chi square, * =p < 0.05, p= [9E−4, 7E−5, 3.94E−05, 1.45E−06], p= [0.353, 0.053,0.522], p =[0.112, 4.94E−16, 9.36E−10], for P6/ +, P6/Kir, P9/Kir, and 25 °C vs 30,35,40 °C, respectively; n.s.= not significant, *p* > 0.05. *N* = 20–44 flies). **f** Schematic of the full circuit described here. Anatomical abbreviations: LH = lateral horn, VLP = ventrolateral protocerebrum, Tct = tectulum, NTct = neck t., WTct = wing t., HTct = halthere t., IntTct = Intermediate t., LTct = lower t., ANm = abdominal neuromere, AMNP = accessory mesothoracic neuropil, IntNP(T1-3) = intermediate neuropil thoracic neuromere 1-3, MB = mushroom body, AL = antennal lobe, NC= neck connective. Source data are provided as a source data file.

---

information. Indeed, in the olfactory system, derivative responses to changes in odorant concentration have been observed and modeled both at the periphery[38–40] and in higher-order neurons[4,5,8–10], where, amongst other functions, they have been suggested to emphasize chemical gradients to facilitate chemotaxis[4,5,8,10,40]; a role conceptually similar to the extrapolation of temperature gradients we show here.

Our results allow us to assign a specific function to ON transients in the context of heat responses in *Drosophila*. More broadly, we suggest that ON transients may represent a general mechanism to systematically anticipate salient conditions and prepare appropriate responses. This could be an expedient strategy to process information about stimuli early on, even before information propagates across a sensory circuit, and could be beneficial when natural stimuli may have a predictable temporal structure and/or when the animal needs to react rapidly, for example, in the case of a looming dangerous temperature.

Here we have shown that, unlike other second-order thermosensory projection neuron cell types, TPN-IIIs selectively relay ON transients independent of absolute temperature and of the direction of change (heating or cooling). We also confirmed that TPN-IIIs receive independent functional input from hot and cold receptors of the antenna (ref. 13,14 and Fig. 1), which leaves the important question of where and how ON transients originate: within sensory neurons of the antenna or at the synapse between TRNs and TPN-IIIs.

Available evidence suggests ON transient likely originate within the TRNs:

(1) Calcium imaging from the terminals of cold TRNs of the antenna has revealed a variety of responses, including persistent calcium transients that depended on absolute temperature and transient calcium spikes in response to cooling, largely independent of absolute temperature[15].

(2)  Patch-clamp recordings from a cold-sensitive cell type of the larval body wall (class III neurons) demonstrated both transient and steady state responses, leading to the suggestion that these cells can encode the rate of cooling by a peak of the firing rate (ON response) and the magnitude of cold temperature by the rate of steady spiking activity[41].

(3)  Sharp-electrode recording of ensemble activity within the shaft of the arista (the compartment that contains the cilia of hot and cold aristal TRNs) demonstrated "spiking" responses to heating or cooling independent of absolute temperature ("phasic" responses[42]).

While we have been so far unable to directly confirm this by patch-clamp recordings from TRNs of the antenna, our hypothesis is that TRN responses contain information about absolute temperature as well and temperature change, the latter in the form of transient ON responses; similar to mammalian thermosensory neurons[43] and larval cold cells[41]. TPN-IIIs selectively extract heating and cooling ON transients from the activity of the antennal TRNs and relay them to the TLHONs for further processing.

Interestingly, while we understand how thermosensory neurons respond to absolute temperature (e.g., by expression of TRP channels gated by temperature at specific thresholds), the biophysical mechanisms that allow TRNs to produce heating and cooling responses independent of absolute temperature remains a mystery. This question merits becoming the subject of direct investigation, as our work now suggests ON responses may have a universally important function in thermosensory coding (and beyond).

## Methods

### Fly strains
*Drosophila melanogaster* strains were reared on cornmeal agar medium at room temperature (-23 °C). Stocks were obtained from Bloomington Drosophila Stock Center (BDSC) or Vienna Drosophila Resource Center (VDRC). The following stocks were used: R22C06-Gal4 (BDSC 48974), R22C06-LexA[27], Gr28Bd-LexA[28], VT061933-Gal4 (VRDC 204294), UAS-Kir2.1 (BDSC:6595), 10xUAS-IVS-mCD8::GFP (BDSC 32186), 13xLexAop-CD2::GFP[44], UAS-DenMark, UAS-sytGFP (BDSC 33064), Dilp2-LexA (N. Yapici Lab), Aop-Kir2.1 (B. Dickson Lab), Aop-P2X2 (BDSC 76030), UAS-GCaMP6m (BDSC 42748), tsh-GAL80[45], DNg06-spGal4 (BDSC 86755), UAS-CsChrimson (BDSC 55136), Aop-CsChrimson (BDSC 55138), VT019018 AD; VT017411-GAL4.DBD (BDSC 75885), VT023490 AD; R38F04-GAL4.DBD (BDSC 75903), Orco[2] (BDSC 23130), wildtype Canton-S (Gallio Lab), MB247DsRed[46], UAS-C3PA-GFP[47]. A full description of genotypes used in each figure can be found in Supplementary Table 4.

### Generation of transgenic flies
To create R22C06-Gal4.AD and VT040053-GAL4.DBD transgenic flies, the corresponding promoter regions were amplified by PCR from *Drosophila* genomic DNA using the primers indicated in FlyLight and Vienna Drosophila Resource Center (VDRC) websites respectively. Entry vectors containing the PCR fragments were generated with the pCR™8/GW/TOPO™ TA Cloning Kit (ThermoFisher Scientific, Cat. # K250020), the inserts were sequenced and moved into pBPp65ADZp-pUw vector (Addgene, # 26234) and BPZpGAL4DBDUw (Addgene, #26233) by LR recombination (ThermoFisher Scientific, Cat #11791100). The resulting expression clones were injected into fly embryos by BestGene Inc.

### Fluorescence microscopy
Two-photon imaging of GFP-labeled neurons was performed on a Prairie Ultima two-photon microscope with a Coherent Chameleon Ti:Sapphire laser tuned to 945 nm, GaAsP PMTs and an Olympus 40 × 0.9NA water immersion objective at 512×512 pixel resolution and 1X or 2X optical zoom. Confocal imaging of immunostained brains was performed on a Zeiss LSM 510 confocal microscope equipped with Argon 450–530 nm, Helium-Neon 543 nm, and Helium-Neon 633 nm lasers and a Zeiss LCI Plan-Neofluar/0.8 DIC Imm Corr 25× objective at 512 × 512 pixel resolution. Maximum projections were obtained from stacks taken at 1 μm steps. Images were processed in Fiji.

### Characterization of Gal4 drivers
A TPN-III driver was generated using VT040053-Gal4.DBD and 22C06-AD-Gal4 split-Gal4 hemi-drivers. The intersection of these hemi-drivers is exclusively active in 4–5 cells per brain hemispheres. TLHON candidate driver lines were identified by visually cross-referencing PA-GFP images to confocal stacks provided in VDRC (https://www.virtualflybrain.org/about/) and FlyLight (https://flweb.janelia.org/cgi-bin/flew.cgi) databases. In single brain hemisphere, VT061933-Gal4 driver is active in the 2 LHPV2g (TLHON) neurons, 4–6 insulin-producing cells (Dilp2-expressing), and weakly in 2 DNg06 descending neurons[30]. Potential confounding effects on behavior due to off-target expression were tested by silencing each cell type using independent drivers (see supplementary material).

### Photo-activation of PA-GFP
To visualize potential postsynaptic partners of TPN-IIIs we photo-activated 1–3 day old flies expressing photo-activatable GFP pan-neuronally under the control of neuronal Synaptorbrevin-Gal4 using a 2-photon microscope[13]. A 3D ROI volume was defined encompassing TPN-III axons (labeled with GFP) within the ventral Lateral Horn to target for photo-activation while the 2-photon laser was tuned to 945 nm (a wavelength for which PA-GFP is not activated). The laser was then tuned to 720 nm (the activation wavelength) and the ROI volume was scanned in 3 μm steps at low power (10–30 mW measured at the back aperture of the objective) 40 times interleaved by 30 s wait periods to allow for photo-conversion and diffusion of the PA-GFP. To generate 2-photon images for analysis, the laser was then re-tuned to 945 nm and a larger field of view was imaged to encompass labeled cells.

### Functional connectivity studies
*For chemogenetic experiments*, ectopic expression of the purinergic receptor, P2X2, in TPN-III was used to determine functional connectivity with TLHONs. In 2–5 days old flies, 22C06 LexA, AOP-P2X2 was expressed presynaptically (in TPN-III), while VT061933 Gal4 driving UAS-GCaMP6m was expressed postsynaptically. To selectively activate TPN-III neurons, ATP (25 mM dissolved in AHL (see below)) was briefly ( ~ 500 ms) puffed nearby TPN-III somas. Alexa Fluor 594 (50 μM, ThermoFisher Scientific Cat. # A10438) was included to visualize the puff. Ca²⁺ imaging was performed as described in AHL continuously perfused with 95% $O_2$ 5% $CO_2$. Temperature was monitored and sampled at 1 KHz to control for thermosensory responses. Images were acquired at 256 × 256 pixel resolution at a rate of 4 Hz on a Prairie Ultima two-photon microscope with a Coherent Chameleon Ti:Sapphire laser tuned to 945 nm. *For optogenetic mapping*, all-*trans* retinal powder (RET, Millipore Sigma, Cat. # R2500) was dissolved in ethanol (Fisher Scientific, Cat. #BP2818500) to prepare a 100 mM stock solution. 1 mL of stock was then mixed with 250 mL of molasses and cornmeal medium to produce 400 μM all-*trans* retinal food. To optogenetically activate thermosensory neurons, HC-Gal4 or CC-Gal4 UAS-CsChrimson flies were crossed to flies in which TPN-III was labeled with GFP. Crosses were set on food laced with all-*trans* RET, covered in foil and placed in a DigiTherm incubator (Trikinetics). Progeny (2–3 days old) for experimentation were then dissected (as above) and placed in perfusate under the 2-photon microscope for TPN-III recordings. HCs expressing CsChrimson were stimulated with a red LED light (660 nm, Thorlabs M660FP1) placed near the brain with a fiberoptic cable (Thorlabs, M28L01). After establishing a whole cell

recoding in a TPN-III, 5 s light stimuli were triggered using a TTL pulse with pClamp software delivered to the LED driver (Thorlabs, LEDD1B). A one-way ANOVA was used to test for significance between light evoked firing rates in CsChrimson expressing and control (no CsChrimson) flies and significance (*) was defined as $p < 0.05$.

## Connectomics

To identify TPN-IIIs and TLHONs in the connectome, we surveyed the EM Hemibrain[22] using Neuprint version 1.2.1 (neuprint.janelia.org). Candidate TPN-III and TLHON skeletons were validated by cross-referencing to light microscopy images (see supplementary material for IDs). To determine synaptic connectivity between TPN-IIIs, TLHONs, and LHPV2a1 etc., we used Python scripts to query Neuprint. TPN-III connectivity diagram in Fig. 2 was determined by querying for the shortest paths between TPN-IIIs and TLHONs. The resulting list of interneurons was grouped by cell type and the cell types were ranked by synaptic weight. To identify additional thermosensory inputs to TLHONs, we first identified thermosensory receptor neurons by querying for the aristal sensory inputs to known hot and cold projection neurons and cross-referencing these skeleton fragments to light microscopy images of the antennal nerve and PAL. We then used these TRN IDs as starting points to query for neurons between the TRNs and TLHONs, with a maximum of 3 allowed connections. The resulting lists of neurons were grouped by cell type and ranked by synaptic weight (see supplementary material for IDs). To identify olfactory inputs to TLHONs, we first identified all uniglomerular olfactory projection neurons originating from the antennal lobe, and then queried these for synaptic connectivity with TLHONs. To generate a word cloud (Fig. 3b) a "wordcloud" function was used in MATLAB by providing the synaptic weights of identified oPNs for respective olfactory glomeruli (see Supplementary Table 2). The function generates text whose relative size linearly corresponds to the range of input values (i.e., synaptic weights). To identify descending neuron targets of TLHONs, we first made a ranked list of common TLHON outputs (defined as any postsynaptic partners that received ≥ 10 synapses from both TLHON cells). Postsynaptic partners that had strong reciprocal connectivity with TLHONs were excluded. From the list of common TLHON outputs, we searched for direct connectivity with any DNs downstream using a synapse threshold of 10.

## Electrophysiology

Whole-cell patch clamp electrophysiology experiments were performed on 2–3 days old male and female flies. Flies were anaesthetized by brief cold exposure in an ice bath (~0 °C) for ~1 min. Using a dissection microscope (Nikon SMZ1000), a small window in the head cuticle was opened and the underlying perineural sheath was gently removed using fine forceps (Moria Surgical). Brain tissue was exposed and bathed in artificial hemolymph (AHL) solution containing the following (in mM): 103 NaCl, 3 KCl, 26 NaHCO₃, 1 NaH2PO4, 8 trehalose dihydrate, 10 dextrose, 5 TES, 4 MgCl₂, adjusted to 270-275 mOSm. For experiments, 1.5 mM CaCl₂ was included and the solution was continuously bubbled with 95% O₂ 5% CO₂ to pH 7.3 and perfused over the brain at a flow rate of 1–2 mL/min. To target neurons for patching under the 2-photon microscope, Gal4 or LexA lines expressing GFP for neuron targeting were excited at 840 nm and detected using a photomultiplier tube (PMT) through a bandpass filter (490-560 nm) using an Ultima 2-photon laser scanning microscope (Bruker, formerly Prairie Technologies). The microscope is equipped with galvanometers driving a Coherent Chameleon laser and a Dodt detector was used to visualize neural tissue/somata. Images were acquired with an upright Zeiss Examiner.Z1 microscope with a Zeiss W Plan-Apochromat 40 × 0.9 numerical aperture water immersion objective at 512 pixels × 512 pixels resolution using PrairieView software v. 5.2 (Bruker). Current clamp recordings were performed with pipettes pulled (Sutter P-97) using borosilicate capillary tubes (WPI, Cat # 1B150F-4) with open tip

resistances of $20 \pm 3$ MΩ filled with internal solution containing the following (in mM): 140 K-aspartate, 1 KCl, 1 EGTA, 10 HEPES, 4 Mg-ATP, 0.5 Na₃-GTP, pH 7.3, 265 mOsm. To visualize the electrode and fill the cell after recording to confirm GFP co-localization, Alexa Fluor 594 Hydrazide (50 μM; ThermoFisher Scientific Cat. # A10438) was included in the intracellular solution, excited using the 2-photon laser at 840 nm, and detected with a second PMT through a bandpass filter (580–630 nm). Recordings were made using Axopatch 200B patch-clamp amplifier and CV203BU headstage (Axon Instruments), lowpass filtered at 2 KHz, scaled to a 20x output gain, digitized with a Digidata 1320 A, and acquired with Clampex software v.9.2.1.9 (Axon Instruments). *Temperature stimulation:* For temperature stimulation, preparations were continuously perfused with Ca²⁺-containing AHL (as described above). AHL was gravity fed through a 3-way valve (Lee company, part # LHDA1231315H) and flow rate was adjusted through a flow regulator. Following the valve, temperature was precisely regulated through 2 in-line solution heater/coolers (Warner Instruments, Cat. # SC-20) in parallel with by a dual channel bipolar temperature controller (Warner Instruments, Cl-200A). Excess heat produced by each SC-20 Peltier was dissipated through a liquid cooling system (Koolance, Cat. # EXT-1055). For standard temperature changes (Fig. 1), AHL flowed through the "baseline" in-line heater/cooler, while the "target" in-line heater/cooler was set to the desired target temperature. Triggering the 3-way valve using a TTL pulse switched flow from the baseline to the target temperature. To achieve slower temperature changes (Fig. 4), no valve was triggered and instead the temperature of the "baseline" in-line heater/cooler was directly adjusted. To circumvent changes in resistivity and voltage offsets from changing the temperature of the bathing solution, the reference Ag-Cl pellet electrode was placed in an isolated well adjacent to the recording chamber (Warner Instruments, Cat. # RC-24N), filled with identical AHL and connected via a borosilicate capillary tube filled by 2% agar in 3 M KCl. The bath temperature was precisely recorded using a custom Type T thermocouple with an exposed tip (Physitemp, Cat. # T-384A) connected to a thermometer (BAT-12, Physitemp) with an analogue output connected to the digitizer and sampled at 10 kHz. The tip of the thermocouple was threaded through a borosilicate capillary tube and precisely placed near the antennae using a micromanipulator (MP-225, Sutter Instruments).

## Behavioral assays

*Group assays*[12,13]. Groups of 20, 5–8 day old well-fed male flies grown under 12 h light: 12 h dark cycles were tested on an array of individually controlled 1″ × 1″ Peltier tiles covered by thin, disposable, black masking tape (Thorlabs). Circular, 1.8″ (45.72 mm) spaces were laser-cut in an 1/8″ (3.175 mm) acrylic sheet to form individual arenas centered over the intersection of 4 Peltier tiles, with serrated edges to prevent flies from climbing on the arena wall and covered by glass (1.8 mm thick). Calibration was performed before each experiment using a FLIR infrared imaging system and a custom MATLAB script. Video of the flies navigating the chamber was acquired using a Chameleon3 USB camera (FLIR) at 3.75 fps using a custom MATLAB script. In each trial, flies were given a choice between 25 °C (BT) and a test temperature (TT) in diagonally opposed quadrants for 3 min, the spatial configuration of BT/TT quadrants was then reversed for an additional 3 min. The following set of test temperatures (BT/TT) were used: 25°/25 °C, 25°/30 °C, 25°/35 °C, 25°/40 °C. Every change in conditions was interleaved by a brief 30 s step at 33 °C to ensure redistribution of the flies. *Single Fly Navigation Assay.* We used well fed 5–7 day old males reared in a 12 L:12D cycle in a temperature- and humidity-controlled environment (25 °C, 40%RH respectively). Prior to being loaded into the arenas, single males were anesthetized on ice for a few seconds. Fly behavior, and movement were recorded using a Chameleon3 USB camera (FLIR) at 30 Hz. Fly tracking and data analysis were performed using Python (v. 3.8), where the OpenCV package was

used to do basic image processing, namely edge detection and approximating the fly body to an ellipse. To calculate the avoidance index (AI) of single flies to the test temperature, we tracked the centroid position of the ellipse during the trial and used the following equation: AI = [time at BT – time at TT] / total time. The direction of motion was calculated using the heuristic that most fly movement is in the forward direction. Translational and rotational velocities at each frame were calculated using the angle of orientation and centroid of the fly. In addition, velocity was projected along the ellipse axis to derive forward and sideways moving direction velocities. The trajectory of flies within the arena was further segmented to pinpoint behaviors in the boundary region between heating elements. This boundary region started at the 25.5 °C isotherm and extended 5 mm (where the temperature is expected to be stable in all conditions). U-turns were considered as events starting at baseline temperature tiles, followed by an intrusion of the boundary region, and terminating back at baseline temperature quadrants. Crossovers were maneuvers that initiated at baseline temperature regions and ended in test temperature tiles. Crossover-to-turn ratio was simply defined as #U-turns / (#U-turns + #Crossovers). To compare these ratios, we used GLMM (Generalized Linear Mixed Models) with fly ID as a random effect and Wald testing to determine significance (threshold $p = 0.05$) as in[28]. For the analysis of turning direction at the boundary region, we analyzed the relationship between the incoming angle of the fly and the outgoing heading direction after the turn (in 25 °C vs 40 °C conditions). Turns were defined as trajectory segments with a deviation in rotational velocity of at least 45°/s. Positive rotation values corresponded to left turns and negative values to right turns. To calculate the incoming angle, we first defined the starting point of the turn and then stepped back within the fly trajectory until the rotational velocity component changed sign or until there was no longer a monotonic decrease in forward velocity. The angle of the body axis at this location, relative to the isothermal lines at the boundary, was considered as the incoming angle. To test effects on the turning bias after silencing experiments, we used GLMM with approach angle and fly ID as random effects and Wald testing to determine significance (threshold p = 0.05). *Four field preference assays.* For olfactometry, 8–13 day old, well-fed male flies were placed inside a custom-built, open-bottom four-field olfactometer in which the floor is a 4 × 4 array of temperature-controlled Peltier tiles. The olfactometer comprises a four-quadrant chamber in which flies walk freely between four air plumes carrying either an odor or a blank while the floor is held at a constant 25 °C. A central vacuum port in the chamber ceiling draws air into the system through four intake ports. The flow rates across each intake port are controlled by independent flowmeters (Cole-Parmer, FR2A12BVBN-CP) and held at a constant 90 mL/min in each quadrant. Airstreams entering the chamber are odorized by bubbling through vials containing 10 mL of either water (for high solubility, hydrophilic odors) or paraffin oil (for low solubility, hydrophobic odors). Control airstreams were bubbled through water or paraffin oil only. Odors were selected based on the known identify of OPNs targeting TLHONs, as well as known aversive odors. To choose odor concentrations that would produce avoidance of the odorized quadrants, we tested a range of concentrations from low ($10^{-3}$ v/v dilution) to high ($10^{-1}$ v/v dilution). Odor concentrations were selected that either elicited behavioral avoidance in the assay, or were in excess (to compensate for plume dilution) of concentrations associated with behavioral responses reported by others (such as oviposition avoidance). Odor concentrations used (in % v/v): 3-Octanol 1% in paraffin oil; benzaldehyde 1% in paraffin oil; phenylethylamine 1% in water; pyrrolidine 2% in water; geosmin 0.3% in water; 2-oxovaleric acid 1% in water. Phenylacetic acid was diluted to 0.15 g/mL in paraffin oil. For each trial, 1 group of 15–20 flies was ice-anesthetized and then placed inside the chamber under 'standard' control conditions (25 °C, 40% RH). Flies were given 5 min to acclimate to the chamber without air flow. Then, the air flow is switched on in all quadrants, and flies are allowed to navigate between air plumes for 9 min. A video of the flies' positions was acquired with an overhead mounted digital camera (Chameleon3 USB camera) at 3.75-5 frames/second using a MATLAB script, and an Avoidance Index (AI) was calculated by plotting the flies' positions within the chamber frame-by-frame using the following simple formula: AI = (n flies in the blank plume) – (n flies in odorized plume)/ Total n of flies. AIs for each frame were then averaged over the total number of frames. *Humidity Preference Assay.* Humidity experiments were carried out in the same manner as the olfactory experiments, except the odorized air plumes were replaced by either dry (20% RH) or humidified (70% RH) plumes. A DG-4 Dewpoint Generator (Sable Systems) was used to produce 20% RH airstreams, while the room was held at a constant 70% RH. Experiments were performed at 25 °C by an environmentally controlled room. *Temperature Preference Assay.* The temperature 2-choice experiment in the 4-field chamber was carried out in the same way as the olfactometry experiments, except the tiles of the floor were switched between the test temperature (TT) and control temperature (25 °C), alternating the position of the temperature every 3 min as described previously. Airflow in each quadrant was 90 mL/min during the temperature choice assay and the avoidance index was calculated using AI = (n flies in 25 °C) – (n flies in TT)/Total n of flies. All video processing was done using custom MATLAB scripts.

## Vehicle modelling

A Braitenberg vehicle[48] that models fly navigation within a thermal landscape in a realistic virtual arena has been described previously[28]. It consists of equations for the vehicle's position and direction

$$\begin{bmatrix} x' \\ y' \\ \theta' \end{bmatrix} = \begin{bmatrix} \frac{1}{2}(v_L + v_R)\cos(\theta) \\ \frac{1}{2}(v_L + v_R)\sin(\theta) \\ \frac{1}{d}(v_R - v_L) \end{bmatrix} \tag{1}$$

where $x$ and $y$ are the position of the vehicle's centroid, $\theta$ is the vehicle's orientation, and $d$ is the distance between the two wheels (set to 0.75 mm, reasonably close to a fly's width). Here $v_L$ and $v_R$ are the speeds of the left and right motor, given by the equations:

$$\begin{aligned} v_L &= w_I h(s_L) + w_C h(s_R) + v_0 + \gamma \\ v_R &= w_C h(s_L) + w_I h(s_R) + v_0 - \gamma \end{aligned} \tag{2}$$

where $w_I, w_C$ refer to the ipsilateral and contralateral weights, respectively, $v_0$ is a base speed (set to 5 mm/s), $\gamma$ is a motor noise term, and $s_L, s_R$ refer to the temperature input at the left and right sensors. In addition, the signals $s_L$ and $s_R$ are 0.3 mm apart at the front of the vehicle and sample the spatially dependent temperature profile in the arena[28]. These signals are passed through a saturating nonlinearity:

$$h(s) = \frac{1}{1 + \exp(-as + b)}, \tag{3}$$

where $s = \{s_L, s_R\}$. The thermosensory input is additively obscured by colored (Ornstein-Uhlenbeck) noise,

$$\tau_s d\varepsilon = -\varepsilon dt + \sigma_s dW, \tag{4}$$

where $\tau_s$ is a time constant for the process, $\sigma_s$ is an amplitude parameter, and $dW$ is the Wiener process. Separate noise processes are used for both left and right inputs, and integration was performed using the Euler-Maruyama method. The left and right sensory inputs are then given by

$$\begin{aligned} s_L &= s_{L0} + \varepsilon_L \\ s_R &= s_{R0} + \varepsilon_R \end{aligned} \tag{5}$$

where $s_{L0}$ and $s_{R0}$ are the true sensor temperatures and $\varepsilon_L$ and $\varepsilon_R$ are the noise terms. The motor noise $\gamma$ is modeled similarly,

$$\tau_m \, d\gamma = -\gamma dt + \sigma_m \, dW \qquad (6)$$

with time constant $\tau_m$ and amplitude $\sigma_m$.

Starting from this basic formulation, we designed a simple filter inspired by a firing-rate model, constructed by modifying a more detailed model of an adapting synapse[49]. For the simplified model we assume an adaptation variable $p$ that evolves according to

$$\tau_p \frac{dp}{dt} = \frac{r_0}{r} - p \qquad (7)$$

where $r(t)$ is the firing rate of the input, and the overall output signal is $g = g_0 p r$. This adaptation model clearly needs modification at small input firing rates, but since here the input rate never becomes zero this detail can be safely ignored. This model possesses a number of properties that made it desirable for detecting changes in the input. In steady state (where we assume constant input $r$), $p = r_0/r$ and thus $g = pr = r_0 g_0$, which is independent of the input $r$. Since we will add an additional weight to this signal, we will take $r_0 = g_0 = 1$ without loss of generality. Another key property is the transient response of $g$ to a fast-changing input. If the input is a step-function $r(t)$ from a smaller input $r_1$ to a larger input $r_2$, since the value of $p$ takes time $\tau_p$ to change appreciably, the immediate change in $g$ follows $\Delta g = p\Delta r = (r_2 - r_1)/r_1$, i.e., the well-known Weber-Fechner law.

We modify the Braitenberg vehicle model to include change signals $g_{\{L,R\}}$. In particular, we modify the left and right motor equations to become:

$$\begin{aligned} v_L &= w_I[h(s_L) + m|g_L - 1|] + w_C[h(s_R) + m|g_R - 1|] + v_0 + \gamma, \\ v_R &= w_C[h(s_L) + m|g_L - 1|] + w_I[h(s_R) + m|g_R - 1|] + v_0 - \gamma, \end{aligned} \qquad (8)$$

where the pre-synaptic inputs to the change signals $g_{\{L,R\}}$ are $r_{\{L,R\}}(t) = h(s_{\{L,R\}}) - h(24°C)$ (we take $24°C$ as the reference temperature since we require $r > 0$). In the above the absolute value of the deviation of the change signals from 1 is used to drive the motors. This allows a single change signal to produce responses to both heating and cooling. A more realistic model should involve two separate change signals, one for increasing and another for decreasing temperatures (representing independent hot and cold TRN input). Overall, the addition of the change signal allows the vehicle to respond to both temperature and the change in temperature.

## Evolutionary optimization of vehicles

Multi-objective optimization of the vehicles was performed via an evolutionary strategy using a modified version of the Non-dominated Sorting Genetic Algorithm II (NSGA-II) method (Abouzeid and Kath, Society for Neuroscience Annual Meeting, DD38; 2019). We evolved the vehicles to optimize five objectives (the squared error between the performance of the vehicle and fly for each objective): (1) the avoidance index and (2) turn-cross ratio at each of the three test temperatures, (3) the probability of a left/right turn given antennal temperature difference at turn start, (4) the "spontaneous" rate of turns per distance walked, (5) probability of early turns. Optimization over a 10-dimensional parameter space $z = \{w_I, w_C, a, b, \tau_s, \sigma_s, \tau_m, \sigma_m, \tau_p, m\}$ was performed. At each generation, individuals from the previous round were "crossed" or "mutated" with probability 0.5. Vehicles were evaluated based on 200 trial simulations (comprised of 50 simulations each of 25/25 °C, 25/30 °C, 25/35 °C, and 25/40 °C). In the same vein as NSGA-II, individuals are first sorted based on dominance. Rather than computing the crowding distances at this step, the errors are first projected into $(N+1)$-dimensional space using stereographic projection (Abouzeid and Kath 2019, see above), and then the crowding distances are calculated. This ensures that errors close to the origin are spread further apart, allowing for better exploration of the Pareto front near the origin. A running Pareto front was stored and updated following each generation. After 500 generations of evolution (each with 112 individuals), we observed strong convergence of the error in each performance criterion among members of the Pareto front. Following evolution, we compared the best performing vehicles in all 5 criteria (defined to be members of the Pareto front with error no >2X the median error for any objective). The top performing vehicle was chosen from these based on a procedure thresholding the original 4 objectives, and sorting based on the 5th: early turns. Note that all 1056 of our best vehicles performed similarly.

## Quantification and statistical analysis

**Detection of action potentials.** Membrane potential recordings were made in current clamp mode sampled at 10 kHz. Data were analyzed offline using Axograph and custom scripts in Igor Pro and MATLAB. Action potentials (spikes) were detected using custom scripts in Igor Pro using Neuromatic v2.6i plug-in. A first derivative transformation was performed on the membrane potential trace defining dV/dt. Then a constant dV/dt threshold was defined (for an individual neuron)– peaks above the threshold defined a single spike Peristimulus time histograms (PTH) of firing rate were made by binning detected spikes in 1 s bins, defining spikes/s (Hz). Traces depicting mean and line and shading indicate the mean firing rate (Hz) ± SEM. For the dose-response curves of TLHON firing rate vs. temperature derivative, spikes were binned into 25 ms bins, and the spike count per bin was used to calculate the firing rate. For the corresponding stimulus sweep, the peak value of the temperature derivative, dT/dt, was detected, and a time interval where dT/dt was at least 90% of its maximum or minimum was defined. The firing rate within this 90% sampling interval was detected and averaged across the sampling interval to give the cell's firing rate at the time when the rate of temperature change was fastest. The pre-stimulus baseline firing rate was defined as the mean firing rate in the 5 s immediately preceding the stimulus onset. Temperature recordings were digitized at 10 kHz and smoothed with a 1000 point span. Smoothed traces were then differentiated and smoothed with a 10,000 point span to generate temperature rate traces where the line and shading indicate the mean rate (°C/s) ± SEM. To test for a significant relationship between temperature and steady state firing rate for TLHONs (Fig. 4), a 1-way ANOVA and test for multiple comparisons were used to determine significance; an asterisk means $p < 0.05$. To test for a relationship between the rate of temperature change and firing rate (Fig. 4) for TPN-III and TLHONs, data were fit with linear functions, and shading indicates the 95% CI. For TLHONs, the fit only includes the portion of the graph containing responses (firing rates that that are above the background firing rate). To determine whether slow thermal change (<0.25 °C/s) elicits responses in TLHONs and TPN-IIIs (i.e., increases in spiking rate from baseline, Fig. 4i,j), background subtracted firing rates at peak stimulus were extracted and binned. A 1-sample $t$-test was used to test if increases in firing rates are different from zero (n.s.= not different from zero, asterisk = different from zero; $p < 0.05$).

**2-choice behavior.** To test for differences in behavior between controls and silenced flies, a 2-way ANOVA was used ($p < 0.05$; Figs. 1–3, and Supplementary Fig. 3). Attraction/Avoidance indexes for each genotype were compared by two-way ANOVA, and asterisks denote a statistically significant interaction between the Gal4 or LexA driver and the UAS or LexAop effector (see figure for the specific N of biological replicates). Kolmogorov–Smirnov tests were used to confirm a normally distributed sample. Homogeneity of variance for each data set was confirmed using Levene's test (threshold $p < 0.05$). To test for the role of TLHON in turning behavior for single flies (Fig. 5), the ratios of turns vs crosses was computed and analyzed using a GLMM (Generalized Linear Mixed Models) with fly ID as a random effect and Wald

testing to determine significance (threshold $P = 0.05$). To compare the occurrence of early vs late turns, we used a Chi-square test to determine whether bins contained equal counts ($p < 0.05$). See Supplementary Data 1 for precise statistical information on Figs. 1–5, 7 and Supplementary Fig. 3.

## Reporting summary

Further information on research design is available in the Nature Portfolio Reporting Summary linked to this article.

## Data availability

All data are available in the main text or as supplementary materials. The relevant raw data from each figure (in the main paper and in the Supplementary Information) are provided as a Source Data File. The following databases were used in this study: Flylight (https://www.janelia.org/project-team/flylight),Vienna Drosophila Resource Center (https://shop.vbc.ac.at/vdrc_store/) and neuPrint (https://neuprint.janelia.org/). Source data are provided with this paper.

## Code availability

Mathematical algorithms used in the paper are reported in the "Methods" section. Custom computer code is available from the Gallio Lab GitHub (https://github.com/MarcoGallio/GallioLab).

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

## Acknowledgements

The authors thank Andrew Kuang and Hamin Gil for technical assistance and Hojoon Lee, Dave McLean and members of the Gallio Lab for comments on the paper. The research reported in this publication was supported by NIH grants R01NS086859, R21EY031849 and R21NS130554, a Pew Scholars Program in the Biomedical Sciences and a McKnight Technological Innovations in Neuroscience Awards (to M.G.). The research was supported in part through the computational resources and staff contributions provided for the Quest high performance computing facility at Northwestern University jointly supported by the Office of the Provost, the Office for Research, and Northwestern University Information Technology (to W.L.K.). R.S. is supported by Training Grant in Circadian and Sleep Research, T32HL007909. J.I.L. was supported by NSF research training grant DMS-1547394.

## Author contributions

M.G., G.C.J., and M.H.A. designed the study, analyzed the data, and wrote the paper with critical input from all authors; M.G., R.S., and W.L.K. designed the vehicle model; R.S. implemented it and carried out evolutionary optimization (with critical input from J.I.L. and W.L.K.). D.D.F. and G.C.J. initially identified TPN-IIIs and TLHONs. D.D.F. performed PA-GFP, P2X2 and DenMark experiments. G.C.J. and D.D.F. performed all imaging experiments. G.C.J. designed, performed, and analyzed all group behavior experiments. M.H.A. performed and analyzed all electrophysiology experiments with help from G.C.J. J.M.S. performed and analyzed all single-fly navigation assays with help from R.S.; G.C.J. performed connectomic analyses with assistance from R.S. and J.I.L. A.P. produced and analyzed transgenic lines and provided critical advice and assistance throughout.

## Competing interests

The authors declare no competing interests.

## Additional information

 **16**