## [Peer Review File · Nature Communications]

Rapid threat assessment in the *Drosophila* thermosensory systemREVIEWER COMMENTS

Reviewer #1 (Remarks to the Author):

This manuscript by Jouandet et al. tackles an important question in sensory neuroscience -- how does a sensory circuit extract a specific temporal pattern from dynamic stimuli to influence locomotion? The authors used a combination of experiments and simulations to support the idea that the rate-of-change encoding is essential for early avoidance behavior to a steep temperature increase (or decrease). They also successfully identified the neural pathways involved in this behavior, all the way from the sensory neurons to the descending neurons.

The authors suggested, in their previous studies, a sensorimotor transformation in which the "temperature derivative" is calculated occurs mostly at the level of TRNs, and TPNIIs develop "broadened derivative tuning" by integrating signals from cold and hot TRNs. This paper focuses on a specific set of lateral horn output neurons postsynaptic to TRNIIs, which appear to exhibit similar "derivative tuning", with a non-linear thresholding process (as shown in Figure 4g-h). These derivative signals ultimately influence movement through DNp06.

Overall, the manuscript is well-written and based on cutting-edge experimental and analytical methods. Experimental techniques include behavioral genetics, calcium imaging, and whole-cell patch clamp recording. The experimental data were analyzed rigorously and presented clearly. In terms of thermosensation, no other research groups currently have the ability to conduct these types of experiments and analyses.

I do have a few points of criticism, though, and some of them might require additional experiments to address.

*major points

1. In most sensory systems, such as vision or olfaction, there exist substantial transformations between subsequent neural layers. The transformation can be either spatial or temporal, or both. This paper reports the role of the third-order sensory neurons in the *Drosophila* thermosensory system alongside with that of the presynaptic second-order neurons. Although the anatomical and functional connectivity is demonstrated convincingly, it is not clear what processing takes place between TPNIIs and TLHONs and how it benefits the thermotaxis behavior of the animal. except for the threshold change. The key difference presented in the paper is (i) the higher derivative threshold of TLHONs than that of TPNIIs, and (ii) the steady-state activity level observed in TLHONs for different baseline temperatures, but not in TPNIIs. It is however not clear what the behavioral consequences of these differences are. I thus ask for additional analyses and experiments for TPNIIs, so as to help assessing the transformation more clearly.

a. The derivative encoding in TPNIIs should be analyzed in the same fashion as in Figure 4e. That is, the data in Figure 1k should be also overlaid with the derivative of the temperature signal. Note that the significant advancement of peak neural responses was observed previously between the second and third-order neurons of the olfactory system in *Drosophila* (Bhandawatt 2007 and Kim et al 2015).

b. If the thresholding is the only processing that takes place between TPNIIs and TLHONs, I can imagine the inactivation phenotype should be comparable between the two cell types, except in the scenario where the temperature rises slowly below the threshold of TLHONs. Furthermore, the "steady state firing rate" is more variable in TLHONs than TPNIIs, which indicates that TLHONs integrate the derivative signal from TPNIIs with the amplitude signal from some other thermosensory neurons. In such a case, it is reasonable to think that the behavioral phenotype upon inactivation might be even stronger for TLHONs than TPNIIs. However, the inactivation experiments with TPNIIs and TLHONs on the group assay appear to show a stronger phenotype in TPNIIs than in TLHONs. I thus think that it is important to conduct the single-fly navigation assay with TPNIIs>Kir flies. This will provide valuable insights into the role of TPNIIs in the early turn responses and shed light on the sensory transformation occurring between TPNIIs and TLHONs.

c. TLHON inactivation phenotype with the single fly assay might be due to partial silencing caused by a weak expression level of the split-GAL4 driver line. In the supplemental Figure 1, I found the expression level of TLHON-GAL4 is relatively weak in TLHONs than in DILP2 cells. To make sure that TLHONs are silenced completely, I would like to suggest performing patch clamp recording on TLHON cells that express Kir2.1.

2. The authors argue that TLHONs carry the “alarm” signals that are weighed more strongly over trials. To support this argument, the change in the probability of “early turns” should be analyzed over trials, as in Figure 6k,m in Simoes et al., 2021. I believe that the change in the probably should be compared between the control and TLHON-GAL4>Kir flies.

3. I wonder whether the olfaction experiments in Figure 3 are essential for the understanding of the main premise of the paper. I recommend most parts of Figure 3 be presented as supplementary figure.

*minor points

1. Line#306 (and Abstract). alarm vs. extrapolation → Extrapolation is a highly specific mathematical term. To this reviewer, “alarm” seems to be a more appropriate choice of a word here. Could you explain why you use the term “extrapolation” here? If “extrapolation” is to be used, I think that a more quantitative role of the derivative signal needs to be demonstrated experimentally. e.g., the rate of temperature rise being 1oC vs. 2oC should result in different physiological or behavioral responses.

2. Could you add a paragraph in Discussion session describing a potential neural circuit in which the “learning” may take place? e.g., a place in which both the derivative and amplitude signals can converge, and the weight for the derivative signals is enhanced over trials.

4. Line#27. “either heating or cooling” should change to “both heating and cooling”

5. Line#51. Please indicate the number of TPNIII cells marked by the original GAL4 lines (VT040053, R22C06).

6. Line#162. a fly it is to turn away. it appears that “it” should be dropped here

7. Line#162. the flies uses → the flies use

8. Line#248. the majority of these DNs has been → the majority of these DNs have been

9. Line#304. The “derivative encoding” in the Drosophila olfactory system needs a bit more explanation. The references given in this paper deal with the temporal encoding in ORNs, corresponding to that in TRNs. What may need explanations is that there exists another layer of temporal processing between ORNs and PNs. Two papers reporting this phenomenon are as follows.

- Bhandawat, V., Olsen, S. R., Gouwens, N. W., Schlieff, M. L. & Wilson, R. I. Sensory processing in the Drosophila antennal lobe increases reliability and separability of ensemble odor representations. *Nature Neuroscience* 10, 1474–1482 (2007).
- Kim, A. J., Lazar, A. A. & Slutskiy, Y. B. Projection neurons in Drosophila antennal lobes signal the acceleration of odor concentrations. *eLife* 4, 1474 (2015).

These papers experimentally demonstrated that there exists another layer of derivative computation between OSNs and PNs, making the onset of the odor stimulus further advanced in

time. The second paper even proposed a model in which the PN output can be predicted by “a double derivative” of odor concentrations, which corresponds to “acceleration” computation. Because this manuscript deals with the second and third-order sensory neurons, these papers may be worth being mentioned and compared with the thermosensory processing reported in this paper.

10. regression method in Figure 5g,h seems a bit arbitrary. It might be important to perform non-parametric regression here. Either compare the existing regression results with those from a non-parametric method or perform parametric regression with multiple functions, and show that the sigmoid provides the least amount of regression error.

11. The number of samples in Figure 5g,h does not match the number of flies or cells in the legend. If samples are drawn from different cells with each cell providing a different number of samples, there exists a problem of bias between different flies. I think that it is statically more appropriate to calculate an average value for each fly and perform the regression on that data. Or, at least use the same number of samples for each fly.

11. In figures, the font style for labels and legends seems often inconsistent. e.g., In Figure 4, the authors use “rate”, “Rate”, “firing rate”, “temperature”, “temp”, “Temp.”, and “peak rate (italicized)”. I would recommend using the same font style throughout all the figures for the labels. Also, it was hard to understand the meaning of the label “mV” in the top right subfigures in Figure 4e. If it was supposed to indicate the unit of the trace, it’s already marked on the y label. In Figure 4j, I would also recommend indicating y labels clearly.

12. At the end of the legend for Figure 5, P30 and P35 are used without being defined. In this case, the author may consider just using “ $p < 0.05$ ”, without bothering to introduce new terms. In Figure 4e, DiffT also seems to require a definition.

13. Line#216, in response to slow-moving looming stimuli \diamond in response to translating or approaching visual objects.

14. In Figure 5e, the color of the legend for fly speed does not match with the color of the fly traces.

15. In Figure 7, subsection f is omitted from the legend. Either include f as part of subsection e, or correct the legend.

Reviewer #2 (Remarks to the Author):

In this manuscript, Jouandet et al identify interneurons in the fly’s thermosensory processing pathway that respond to changes in temperature using brief electrophysiological depolarization (on-transients) and enable rapid behavioral responses such as escape turns.

The discoveries are interesting and rigorously demonstrated. Only textual suggestions for clarification are needed.

In the introduction, the goal of comparing this mechanism to other sensory modalities is admirable but it was too abbreviated to unpack. Sensory systems use information differently for directed escape, navigation, and discrimination tasks. The strongest finding here seems to be detection of temperature change – in either direction – can be used for rapid response, especially at condition boundaries. Which is actually the best analogous system to compare mechanisms and potential roles for on-transients? I would have thought it would be the visual system and suggest expanding

the exploration of coding similarities. What about olfaction? Are odor off-transients used similarly to stay on a gradient during navigation? Is speed key there? The sensory neurons themselves are thought to be the derivative-detectors in that system.

The figures are very dense. While this is a style choice and maybe needed to present the range of supporting data, a simpler summary figure would make the manuscript more accessible. Since the hope is that a description of this mechanism for information coding may generalize across sensory modalities and model organisms, highlighting the key observations about the circuit and the neural coding in a less dense way is recommended. Possibly a version of Figure 7a that includes the relevant sensory inputs and conceptual layers (as well as the real names) of only the neurons in focus for this processing stream? As is, there are many complex and overlapping schematics (2D and F, etc.) but no simple overview.

Figure 5e and f present the behavioral consequences of silencing the TLHON population very intuitively – but connecting the TLHON effect to that of disrupting the 5-6 TPN-III cells and especially the consequences of disrupting their ability to generate on-transients to behavior was more tenuous. The avoidance index and two choice assay do not speak to the model of early escapes as effectively.

The presence of the on-transients are well documented – indeed, electrophysiology is essential and a technical tour de force – and the connectome circuit analysis is a powerful guide for the behavioral tests, but there is a missing link to show that it is actually the on-transients in these neurons that is how they encode the critical information. Is it possible to disrupt or reproduce the on-transients more specifically and test their effects?

The Braitenberg Vehicle model is an intuitive and appealing addition.

The language about learning is unclear (lines 165, 169, 187, 232). The advantage of rapid response to thermal change is likely innate – why should the flies have to learn that an abrupt change in temperature represents a quadrant boarder? There was no experimental evidence shown that performance changes with experience, so sharpening the language to avoid this confusion would help.

Overall, this is an important contribution to understanding how animals encode information about temperature change and use it to guide efficient behavioral responses.

Reviewer #3 (Remarks to the Author):

This manuscript is to investigate the neural circuit involved in guiding the escape behavior in response to potentially dangerous thermal conditions. The authors have identified a comprehensive neural circuit encompassing sensory input and behavioral output. The following are my suggestions to help improve this manuscript:

1. This neural circuit (TRNs – TPN-IIIs – TLHONs – DNp06) guides the escape behavior. The authors used Kir to silence these neurons and tested their necessity for escape behavior. It is also valuable to explore whether this neural circuit is sufficient to drive escape/avoidance behavior.
2. TPN-IIIs are independently driven by hot- and cold-activated TRNs of the antenna, and the TLHONs are the downstream neurons of TPN-IIIs. However, the authors have only examined the behavioral function of TLHONs in response to sudden temperature increases. It would be valuable to investigate the role of TLHONs when flies encounter a sudden temperature decrease, as TLHONs are also activated by temperature decreases.
3. DNp06 is the output neuron in this circuit. It is crucial to determine whether DNp06 is necessary and sufficient for temperature preference assays. Additionally, evidence for the functional connection between TLHONs and DNp06 is also important.
4. It is recommended that the authors adhere to the *Drosophila* gene nomenclature when referring to genotypes and mutants in both the text and figures of the manuscript.

5. Figure 1: (a) The red and brown colors in Figure 1a are difficult to differentiate. It would be beneficial to enhance the color contrast. (b) The quality of Figure 1b is suboptimal, making it challenging to discern the cell numbers depicted. Consider improving the image quality or providing additional details to help identify the cell numbers more accurately. (k) It is unclear why the authors employed a two-way ANOVA in Figure 1k. It is worth noting that similar figures, such as 2k and 3d, implemented a one-way ANOVA. Please provide a rationale for using different statistical analyses in these similar figures. (k and l) In Figure 1k, the red peak 1 appears smaller than 25 Hz, whereas, in Figure 1l, the red peak 1 appears to be approximately 25 Hz. However, according to the legend, these two values should be the same. Please clarify and explain the discrepancy. (g) It would be beneficial to include a representative figure of CC>Chrimson in Figure 1g to provide a more comprehensive visualization of the experimental results.
6. Figure 2: (e) Please provide additional information about Figure 2e in the legend to enhance understanding of the data presented. (g) To clearly indicate the presence of two cell bodies in Figure 2g, consider using arrows or arrowheads to highlight their locations. (h) The red and green color choices in Figure 2h may pose challenges for color-blind readers. It is advisable to select alternative colors. (j) The labeling of Gal4>P2X2 and Gal4/+ in Figure 2j may be confusing and difficult to understand. It would be helpful if the authors utilized the same labeling as in Figure 2h for consistency and clarity. Additionally, in the legend, if "synapse counts" and the "aggregate # of synapses" refer to the same concept, it is recommended to use consistent terminology throughout the legend.
7. Figures 3d and 2k: How did the authors explain the different results for the two-choice assay between 25 and 35°C in these two figures?
8. Figure 3g: The labeling is not clear. It is recommended to change the labeling of experimental groups to dark grey.
9. Figure 4j: In the top panel, depolarization is observed. It would be beneficial for the authors to provide an example demonstrating that when the temperature change rate is lower than 0.2°C/s, there is no activation observed.
10. Figure 5c, 5g, 7e: The statistical analysis of 25°C vs 40°C is not discussed in the figure legends. Additionally, in Figures 5g and 7e, it is necessary to label the temperatures of each assay (e.g., 25°C vs 30°C, 25°C vs 35°C, and 25°C vs 40°C).
11. In Figure 5d, the authors stated that there "are not significantly different between control and experimental groups." However, the specific statistical method used in this analysis is not mentioned.

We are now submitting a fully revised version of our manuscript for your review. First, let us thank you for your very competent handling of our manuscript and let us thank the reviewers for engaging with our work and for their thorough and supportive reports. As you will see we worked hard on the revision, and **our new manuscript addresses all of the reviewers' comments and incorporates most if not all of their suggestions**. To address their points, we produced **10 new data Figures/Panels** (see below); **3 new data panels are now also incorporated in main Figures**, and **2 are now new Supplementary Figures** submitted with the paper. We have also re-worked the text (and marked the largest edits in yellow), figures and legends, to incorporate the reviewer's suggestions and to make further clarifications based on their comments. We are excited to submit to you a much-improved manuscript.

REVIEWER COMMENTS

Reviewer #1 (Remarks to the Author):

This manuscript by Jouandet et al. tackles an important question in sensory neuroscience -- how does a sensory circuit extract a specific temporal pattern from dynamic stimuli to influence locomotion? The authors used a combination of experiments and simulations to support the idea that the rate-of-change encoding is essential for early avoidance behavior to a steep temperature increase (or decrease). They also successfully identified the neural pathways involved in this behavior, all the way from the sensory neurons to the descending neurons.

The authors suggested, in their previous studies, a sensorimotor transformation in which the "temperature derivative" is calculated occurs mostly at the level of TRNs, and TPNIII develop "broadened derivative tuning" by integrating signals from cold and hot TRNs. This paper focuses on a specific set of lateral horn output neurons postsynaptic to TRNIII, which appear to exhibit similar "derivative tuning", with a non-linear thresholding process (as shown in Figure 4g-h). These derivative signals ultimately influence movement through DNp06.

Overall, the manuscript is well-written and based on cutting-edge experimental and analytical methods. Experimental techniques include behavioral genetics, calcium imaging, and whole-cell patch clamp recording. The experimental data were analyzed rigorously and presented clearly. In terms of thermosensation, no other research groups currently have the ability to conduct these types of experiments and analyses. I do have a few points of criticism, though, and some of them might require additional experiments to address.

We thank this reviewer for their supportive comments.

*major points

1. In most sensory systems, such as vision or olfaction, there exist substantial transformations between subsequent neural layers. The transformation can be either spatial or temporal, or both. This paper reports the role of the third-order sensory neurons in the *Drosophila* thermosensory system alongside with that of the presynaptic second-order neurons. Although the anatomical and functional connectivity is demonstrated convincingly, it is not clear what processing takes place between TPNIII and TLHONs and how it benefits the thermotaxis behavior of the animal. except for the threshold change. The key difference presented in the paper is (i) the higher derivative threshold of TLHONs than that of TPNIII, and (ii) the steady-state activity level observed in TLHONs for different baseline temperatures, but not in TPNIII. It is however not clear what the behavioral consequences of these differences are. I thus ask for additional analyses and experiments for TPNIII, so as to help assessing the transformation more clearly.

a. The derivative encoding in TPNIII should be analyzed in the same fashion as in Figure 4e. That is, the data in Figure 1k should be also overlaid with the derivative of the temperature signal. Note that the significant advancement of peak neural responses was observed previously between the second and third-order neurons of the olfactory system in *Drosophila* (Bhandawatt 2007 and Kim et al 2015).

As requested, we have now analyzed and displayed the responses of TPN-IIIs in the same way as we had reported the responses of TLHONs (New, Supplementary Figure 2, also shown below). The reviewer

will note that we do not see a significant advancement of peak neural responses as the one reported in the olfactory system.

Figure SN. TPN-III ON responses report fast heating or cooling.

Inspired by this comment (and this reviewer's point #9), we also overlaid directly TLHON responses with the first and second derivative of the stimulus (Figure, below). We believe this makes it clear that TLHONs responses peak in correspondence of the derivative of the stimulus and not earlier.

We note that, unlike the olfactory system, the thermosensory system comprises a set of diverse sensory neurons responding to stimuli with different tuning and complex dynamics (see for Example, Alpert et al., 2020). The second- and third-order transformations in this system may perhaps also serve different roles. We are nonetheless grateful to this reviewer for pointing out the potential for this interesting transformation, which may indeed be relevant in this system (we now cite the relevant papers and comment on this in the discussion section, see below).

b. If the thresholding is the only processing that takes place between TPNIIIs and TLHONs, I can imagine the inactivation phenotype should be comparable between the two cell types, except in the scenario where the temperature rises slowly below the threshold of TLHONs. Furthermore, the "steady state firing rate" is more variable in TLHONs than TPNIIIs, which indicates that TLHONs integrate the derivative signal from TPNIIIs with the amplitude signal from some other thermosensory neurons. In such a case, it is reasonable to think that the behavioral phenotype upon inactivation might be even stronger for TLHONs than TPNIIIs. However, the inactivation experiments with TPNIIIs and TLHONs on the group assay appear to show a stronger phenotype in TPN-III flies than in TLHONs. I thus think that it is important to conduct the single-fly navigation assay with TPNIII>Kir flies. This will provide valuable insights into the role of TPNIIIs in the early turn responses and shed light on the sensory transformation occurring between TPNIIIs and TLHONs.

The reviewer is correct in pointing out that some interesting transformation takes place between subsequent neural layers here (their point 1 above). Indeed, we have performed single-fly navigation assays in TPNIII>Kir flies and observed a different/broader phenotype than what we observed in TLHONs, namely **TPN-III>Kir flies move less in our assays than control flies, especially in 2-choice temperature preference experiments** (in 25 $^{\circ}$ vs30 $^{\circ}$ C, 25 $^{\circ}$ vs35 $^{\circ}$ C and 25 $^{\circ}$ vs40 $^{\circ}$ C, as opposed to "mock" 2-choice experiments, 25vs25 $^{\circ}$ C; see Figure below). For example, the average distance travelled at 25vs25 $^{\circ}$ C for TPN-III>Kir flies during an assay is of 107mm vs 133mm and 155mm of the controls, while the average distance travelled at 25vs40 $^{\circ}$ C is only 48mm vs 113mm and 185mm of the controls (importantly less overall movement does not correlate with lower avoidance, as TPN-III>Kir flies produce normal avoidance to 40 $^{\circ}$ C, see Figure 2).

TLHON-silenced flies do not move less than control (in fact they move somewhat more in some experimental conditions, but not the wide distribution of data points). **We interpret this difference in phenotype to mean that additional TPN-III targets, distinct from TLHONs, are likely involved in promoting movement in response to heat exposure in our assay.** Because TPNIII>Kir flies move overall less, their encounters with the border are also fewer so that it proved hard to compare the dynamics specific border trajectories (escape turns etc.) to those of controls. For this reason, we have not attempted to draw any conclusion on the specific effect/s of their silencing on turning behavior. We think the broader phenotype of TPNIII silencing is interesting as it points to additional TPN-III targets involved in locomotion and in the response to heat. We are now actively pursuing these targets. In the context of this paper, we decided to focus on TLHON as they have a narrower/ more specific phenotype which is relevant for navigation.

Distance data was acquired from single-fly 2-choice temperature preference assays. In each experiment, flies navigate in an arena whose floor tiles are set to produce a choice between 25°C or a test temperature (30°, 35° or 40°C as indicated; see methods for details). As a control, overall distance travelled was also recorded during a “mock” experiment in which test temperatures were set at 25°C (i.e., no choice). Asterisks indicate a significant interaction between experimental genotypes and the appropriate controls in 2-way ANOVA ($p < 0.05$; black line = median, box = interquartile range, whiskers = range, dots = individual flies). Lines and p values in the top box denote a difference in means (one side T test, $p < 0.05$).

c. TLHON inactivation phenotype with the single fly assay might be due to partial silencing caused by a weak expression level of the split-GAL4 driver line. In the supplemental Figure 1, I found the expression level of TLHON-GAL4 is relatively weak in TLHONs than in DILP2 cells. To make sure that TLHONs are silenced completely, I would like to suggest performing patch clamp recording on TLHON cells that express Kir2.1.

Thank you for this suggestion. **We now show patch-clamp recordings from TLHON-Kir flies, demonstrating that TLHONs are indeed silenced by Kir** (see below and **New Supplementary Figure 1**). As suggested above, when it comes to the potentially stronger/different phenotype obtained by silencing TPN-III as compared to TLHON, our interpretation is that, since TPN-IIIs target a number of additional cell types besides TLHONs, one or more of these yet-to-be-characterized cell types may also be relevant for thermal preference behavior. In general, we have been cautious when comparing TPN-III and TLHON phenotypes, on the one hand because TLHONs are not the sole target of TPN-IIIs, on the other because additional TPNS exist that carry thermal information in parallel to TPN-IIIs to the LH and other brain areas (including TPN-V, that targets TLHONs in parallel to TPN-IIIs).

2. The authors argue that TLHONs carry the “alarm” signals that are weighed more strongly over trials. To support this argument, the change in the probability of “early turns” should be analyzed over trials, as in Figure 6k,m in Simoes et al., 2021. I believe that the change in the probably should be compared between the control and TLHON-GAL4>Kir flies.

The reviewer alludes to the fact that we have previously shown that flies learn to produce “early turns” (turn early when they encounter the hot gradient) to efficiently avoid noxious/sub-noxious heat (35-40°C). This effect is evident during the course of a trial, when wild type flies initially produce cross-overs or deep turns before resorting more and more frequently to “early” turns (resulting in a negative slope in the fit for max temperature experienced during interactions over time, see **panel j-k here, reproduced from Simoes et al., 2021**). This effect is also evident when completely naïve flies are subjected to consecutive short trials, in this case early turns overall increased as a function of trial number, while a day later the effect was reversed. We described this phenomenon as form of simple learning (Simoes et al., 2021)

In this manuscript, we discuss the possibility that: “... our results support the notion that an ON “derivative” signal such as the one we observe in TLHONs can be flexibly used to modulate the response to temperature change, such that a small heating step may become salient, and induce a significant escape response (an early turn) if external conditions (imminent thermal danger) or previous experience (learning) suggest to the animal this may be an adaptive strategy”.

TLHON-Kir flies produce overall fewer early turns (in favor of more cross-overs and deep turns, see manuscript **Figure 5**), and one additional expectation is that (if indeed the ON signal TLHONs provide is weighed more strongly over border encounters, i.e. is an essential component of learning) upon TLHON silencing early turn probability should not increase over time (the data plotted as in **k** above should no longer be fit by a line with negative slope). Our analysis shows that this is likely the case. Unfortunately, unlike wild type flies (the object of Figure 6k,m in Simoes et al., 2021) line fits for both TLHON-Kir and control flies (Gal4/+ and Kir/+) are not particularly convincing (see **Figure** below. And note that we do not see a phenotypic rescue by running flies over and over, e.g. in multiple subsequent trials, data not shown).

To test more directly the possibility that TLHON responses may become potentiated as a result of learning we also patched TLHONs in completely naïve flies (never exposed to rapid thermal change), and compared their responses to those of “expert” flies, i.e. flies that had performed 5 independent short trials as described above. We so far saw no convincing difference between the groups (see **Figure** below, importantly, we believe this data should be taken with a grain of salt as the flies had to be immobilized and their head cuticle dissected to gain patch-clamp access to the cell, introducing a significant temporal confound and potentially disrupting the brain milieu). In the absence of stronger data, we have decided to merely suggest the *possibility* that these ON transients *could* be used in the context of learned responses, either upstream or as part of a learning signal.

3. I wonder whether the olfaction experiments in Figure 3 are essential for the understanding of the main premise of the paper. I recommend most parts of Figure 3 be presented as supplementary figure.

As we mention in the manuscript, based on the sparse targeting of TLHONs by select TPNs our initial hypothesis was that these cells may be involved in the processing of both aversive olfactory and thermosensory cues. This 4-field olfactometer set up (which took a considerable amount of work to establish) was designed to test olfactory responses and thermosensory responses in the same chamber so that flies would be using the same repertoire of maneuvers (turns etc.) to avoid either heat or aversive odors. The results clearly show that even in this chamber TLHON silencing affects temperature responses but not odor responses (attraction/avoidance). This is an interesting result (and please note that the manuscript does not dwell on this more than seems appropriate). Moreover, we generally prefer to include all results that are described in the paper as main figures (and use supplementary figures for essential controls that do not warrant a lengthy discussion). For these reasons we decided to keep the figure as part of the main paper.

*minor points

1. Line#306 (and Abstract). alarm vs. extrapolation → Extrapolation is a highly specific mathematical term. To this reviewer, “alarm” seems to be a more appropriate choice of a word here. Could you explain why you use the term “extrapolation” here? If “extrapolation” is to be used, I think that a more quantitative role of the derivative signal needs to be demonstrated experimentally. e.g., the rate of temperature rise being 1°C vs. 2°C should result in different physiological or behavioral responses.

The reviewer is correct to point out that we are not suggesting flies estimate a precise future temperature value based on a trend of thermal change they experience, we are only suggesting that rapid thermal change is salient to them because it can quickly lead to *future* potentially deleterious consequences (it is not dangerous *per se* but salient because of its *extrapolated* consequences). When it comes to experimental grounding of this model, we show that TLHONs respond to rapid, but not to slow thermal change (i.e. rapid change elicits a “different physiological response”). We would like to keep the word “extrapolate” as we think it communicates well the message we want to convey. Of course we are open to delete the word if the reviewer (and editor) feel strongly this should be the case, in which case it would be helpful if they could provide simple alternatives (we think for example that “alarm” would be misleading as it normally implies a negative valence).

2. Could you add a paragraph in Discussion session describing a potential neural circuit in which the “learning” may take place? e.g., a place in which both the derivative and amplitude signals can converge, and the weight for the derivative signals is enhanced over trials.

Thank you for this suggestion, this is an object of intense work in our laboratory at the moment. As explained above (Response to point 2), we have tried hard to establish if learning directly leads to potentiation of TLHON activity or output. If not, the derivative signal may be weighed differently at a point downstream (as the reviewer suggests). Unfortunately the results we obtained so far do not yet allow us to rule out any model and therefore we are hesitant to propose specific hypotheses.

4. Line#27. “either heating or cooling” should change to “both heating and cooling”

Thank you for this suggestion. We now corrected this sentence.

5. Line#51. Please indicate the number of TPNIII cells marked by the original GAL4 lines (VT040053, R22C06).

Thank you for this suggestion. We now clearly indicate the number of cells in each line.

6. Line#162. a fly it is to turn away. it appears that “it” should be dropped here

Thank you for this suggestion. We now corrected this sentence.

7. Line#162. the flies uses → the flies use

Thank you for this suggestion. We now corrected this sentence.

8. Line#248. the majority of these DNs has been → the majority of these DNs have been

Thank you for this suggestion. We now corrected this sentence.

9. Line#304. The “derivative encoding” in the *Drosophila* olfactory system needs a bit more explanation. The references given in this paper deal with the temporal encoding in ORNs, corresponding to that in TRNs. What may need explanations is that there exists another layer of temporal processing between ORNs and PNs. Two papers reporting this phenomenon are as follows.

• Bhandawat, V., Olsen, S. R., Gouwens, N. W., Schlieff, M. L. & Wilson, R. I. Sensory processing in the *Drosophila* antennal lobe increases reliability and separability of ensemble odor representations. *Nature Neuroscience* 10, 1474–1482 (2007).

• Kim, A. J., Lazar, A. A. & Slutskiy, Y. B. Projection neurons in *Drosophila* antennal lobes signal the acceleration of odor concentrations. *eLife* 4, 1474 (2015).

These papers experimentally demonstrated that there exists another layer of derivative computation between OSNs and PNs, making the onset of the odor stimulus further advanced in time. The second paper even proposed a model in which the PN output can be predicted by “a double derivative” of odor concentrations, which corresponds to “acceleration” computation. Because this manuscript deals with the second and third-order sensory neurons, these papers may be worth being mentioned and compared with the thermosensory processing reported in this paper.

Thank you for bringing this to our attention. These papers are indeed very relevant background for this work, and we agree that it was an oversight on our part not to discuss them more directly. We now cite these papers and comment on them specifically in the discussion section.

10. regression method in Figure 5g,h seems a bit arbitrary. It might be important to perform non-parametric regression here. Either compare the existing regression results with those from a non-parametric method or perform parametric regression with multiple functions, and show that the sigmoid provides the least amount of regression error.

We agree with this assessment. The point of this panel is to show as clearly as possible that TLHONs have a spiking threshold while TPN-IIIs do not, hence our earlier attempt to fit a sigmoid to TLHON responses. We now use a different strategy: we only fit the part of the plot that contains non-zero responses in **g** to emphasize the threshold. Separately, we show in a new panel (**i,j**, see below) background-subtracted firing rates from 8 cells/6 animals exposed in sequence to a slow and a fast thermal change. Panels **i,j** clearly show that slow rates of temperature change do not produce increases in firing rate above baseline for TLHONs but do so for TPN-IIIs. We thank the reviewer for this point, we believe the figure is much improved as a result of their comment.

11. The number of samples in Figure 5g,h does not match the number of flies or cells in the legend. If samples are drawn from different cells with each cell providing a different number of samples, there exists a problem of bias between different flies. I think that it is statically more appropriate to calculate an average value for each fly and perform the regression on that data. Or, at least use the same number of samples for each fly.

We think that the reviewer is referring to Figure 4g,i (instead of Figure 5). The reviewer is correct that the we included datapoints from different number of cells/flies and that each cell included has a distinct number of datapoints. This is because, unfortunately, while our stimulation set-up produces flow that is largely laminar, there are still small differences in flow rate that result in small differences in the rate of thermal change. We did not want to aggressively bin the data, as we wanted the reader to get a sense of the extent of variability we see by showing the full raw dataset. To directly address the reviewer's concern that unequal samples/cell may introduce bias into the data we now show a **within cell** comparisons of background-subtracted firing rates from 8 cells/6 animals exposed in sequence to slow and a fast thermal change stimuli (**Figure 4 i,j**, see above).

11. In figures, the font style for labels and legends seems often inconsistent. e.g., In Figure 4, the authors use "rate", "Rate", "firing rate", "temperature", "temp", "Temp.", and "peak rate (italicized)". I would recommend using the same font style throughout all the figures for the labels. Also, it was hard to understand the meaning of the label "mV" in the top right subfigures in Figure 4e. If it was supposed to indicate the unit of the trace, it's already marked on the y label. In Figure 4j, I would also recommend indicating y labels clearly.

Thank you for bringing this to our attention. We now corrected the figures to harmonize notations. Note that "mV" on top of Figure 4e is there for symmetry with the labels below.

12. At the end of the legend for Figure 5, P30 and P35 are used without being defined. In this case, the author may consider just using " $p < 0.05$ ", without bothering to introduce new terms. In Figure 4e, DiffT also seems to require a definition.

Thank you for this suggestion. We now corrected the legend and defined DiffT.

13. Line#216, in response to slow-moving looming stimuli \diamond in response to translating or approaching visual objects.

Thank you for this suggestion. We now corrected this sentence as requested.

14. In Figure 5e, the color of the legend for fly speed does not match with the color of the fly traces.

Thank you for bringing this to our attention. We now corrected the figure.

15. In Figure 7, subsection f is omitted from the legend. Either include f as part of subsection e, or correct the legend.

Thank you for bringing this to our attention. We now corrected the figure legend as suggested.

Reviewer #2 (Remarks to the Author):

In this manuscript, Jouandet et al identify interneurons in the fly's thermosensory processing pathway that respond to changes in temperature using brief electrophysiological depolarization (on-transients) and enable rapid behavioral responses such as escape turns.

The discoveries are interesting and rigorously demonstrated. Only textual suggestions for clarification are needed.

In the introduction, the goal of comparing this mechanism to other sensory modalities is admirable but it was

too abbreviated to unpack. Sensory systems use information differently for directed escape, navigation, and discrimination tasks. The strongest finding here seems to be detection of temperature change – in either direction – can be used for rapid response, especially at condition boundaries. Which is actually the best analogous system to compare mechanisms and potential roles for on-transients? I would have thought it would be the visual system and suggest expanding the exploration of coding similarities. What about olfaction? Are odor off-transients used similarly to stay on a gradient during navigation? Is speed key there? The sensory neurons themselves are thought to be the derivative-detectors in that system.

Thank you for this suggestion. We now re-wrote and expanded the introduction to better frame ON responses in the context of what we think are likely universally well-known ON responses in vision and touch. We discuss similarities with the olfactory system in more detail in the discussion section.

The figures are very dense. While this is a style choice and maybe needed to present the range of supporting data, a simpler summary figure would make the manuscript more accessible. Since the hope is that a description of this mechanism for information coding may generalize across sensory modalities and model organisms, highlighting the key observations about the circuit and the neural coding in a less dense way is recommended. Possibly a version of Figure 7a that includes the relevant sensory inputs and conceptual layers (as well as the real names) of only the neurons in focus for this processing stream? As is, there are many complex and overlapping schematics (2D and F, etc.) but no simple overview.

Thank you for this excellent suggestion. We now added a summary panel to Figure 7 intended to be an easily accessible summary.

Figure 5e and f present the behavioral consequences of silencing the TLHON population very intuitively – but connecting the TLHON effect to that of disrupting the 5-6 TPN-III cells and especially the consequences of disrupting their ability to generate on-transients to behavior was more tenuous. The avoidance index and two choice assay do not speak to the model of early escapes as effectively.

We agree with this assessment. When it comes to the phenotype of silencing TPN-IIIs, as discussed above (points 1 and 2 raised by Reviewer 1) the effect we observe from high-resolution single fly assays is that TPN-III-silenced flies move very little in our 2-choice thermal preference experiments. This phenotype is quite different from that of TLHONs and suggests that TPN-IIIs likely target additional circuits that regulate motility in our assays, distinct from TLHONs. As we suggest above (see answer to Reviewer #1) we do know that TLHONs are not the sole target of TPN-IIIs and that additional TPNs target TLHONs in parallel to TPN-IIIs.

The presence of the on-transients are well documented – indeed, electrophysiology is essential and a technical tour de force – and the connectome circuit analysis is a powerful guide for the behavioral tests, but there is a missing link to show that it is actually the on-transients in these neurons that is how they encode the critical information. Is it possible to disrupt or reproduce the on-transients more specifically and test their effects?

The activity of TLHON is characterized by a small baseline modulation by absolute temperature (cells fire at ~5Hz at stable 20°C, but are essentially silent at 25° and 30°C) and by prominent, absolute-temperature independent ON transients that report thermal change (manuscript **Figure 4c,d**). The behavioral phenotypes that result from TLHON silencing span the range of 25vs10°C - 25vs35°C (in 2-choice assays, manuscript **Figure 2k**) and are therefore likely to reflect a loss of their ON responses.

We agree that it would have been powerful to be able to directly demonstrate the significance of ON transients. The ideal experiment here would have been to produce an artificial ON response as the fly is approaching a non-dangerous thermal boundary (e.g. 25vs30, low probability of escape turns) and to assess if this ON signal alone is sufficient to bias turning probability. Unfortunately this experiment is not currently technically within our capabilities as it would require a closed loop optogenetic system that could target a freely moving fly as it is approaching the gradient (assuming we can produce a sharp optogenetic stimulus that closely reproduces an ON transient in a moving fly, which may not be trivial based on our experience with optogenetic activation of neurons).

The Braitenberg Vehicle model is an intuitive and appealing addition.

Thank you very much for this comment.

The language about learning is unclear (lines 165, 169, 187, 232). The advantage of rapid response to thermal change is likely innate – why should the flies have to learn that an abrupt change in temperature represents a quadrant boarder? There was no experimental evidence shown that performance changes with experience, so sharpening the language to avoid this confusion would help.

We apologize for this confusion. We have previously shown that, when encountering a steep thermal gradient, flies quickly decide if to either cross-over into the hot quadrant or to turn back, and that the probability of turning back is a function of the temperatures they encounter -this is indeed likely a simple innate response. But we have also shown that flies learn remarkably rapidly to perform heat escape maneuvers, so that even during a 3 minute trial, at each encounter they progressively turn earlier and earlier within the gradient and effectively minimize their exposure to heat (this conclusion is based on a number of results detailed in Simoes et al 2021). *We have tried to clarify the relevant sections of the paper to make this clear.*

Overall, this is an important contribution to understanding how animals encode information about temperature change and use it to guide efficient behavioral responses.

Thank you very much for this supportive comment, which was much appreciated by all authors.

Reviewer #3 (Remarks to the Author):

This manuscript is to investigate the neural circuit involved in guiding the escape behavior in response to potentially dangerous thermal conditions. The authors have identified a comprehensive neural circuit encompassing sensory input and behavioral output. The following are my suggestions to help improve this manuscript:

1. This neural circuit (TRNs – TPN-IIIIs – TLHONs – DNs) guides the escape behavior. The authors used Kir to silence these neurons and tested their necessity for escape behavior. It is also valuable to explore whether this neural circuit is sufficient to drive escape/avoidance behavior.

As suggested above (see response to Reviewer 2), we agree that it would have been powerful to be able to directly demonstrate the significance of ON transients. Unfortunately closed-loop experiments such as the ones that would have been required for this are not currently technically within our capabilities. We did try “generic” optogenetic activation of TLHONs (using CsChrimson), and we can conclude that this produces aversion to the optogenetic stimulus (see our unpublished data **Figure** below). In our experience this outcome is quite common, hence we do not consider this a useful datapoint to better understand the function of TLHONs.

(unpublished data by M. Capek, Gallio Lab 2023)

2. TPN-IIIIs are independently driven by hot- and cold-activated TRNs of the antenna, and the TLHONs are the downstream neurons of TPN-IIIIs. However, the authors have only examined the behavioral function of TLHONs in response to sudden temperature increases. It would be valuable to investigate the role of TLHONs when flies encounter a sudden temperature decrease, as TLHONs are also activated by temperature decreases.

This is an excellent suggestion but unfortunately again a difficult point to address experimentally. In our 2-choice assays, as the flies move away from a hot quadrant, they are exposed to rapid cooling. This cooling would be expected to carry a positive valence (as it represents potential respite from heat) but would still activate TLHONs (and therefore should be perceived as a “salient” temperature change). What specific phenotype do we expect from TLHON’s silencing if indeed this manipulation reduces this thermal change’s salience? One possibility is that the hot-cool trajectories may prove less linear in TLHON-Kir flies than in controls. In order to investigate this in detail, we analyzed trajectories that started in the hot quadrant and ended in the 25°C quadrant (corresponding to rapid cooling, see **Figure** below). We analyzed potential differences in Arc/Chord ratio (a measure of trajectory directedness) as well as the average angular velocity of such trajectories (in case flies turn side to side rather than moving in a straight line). Only one of these measures showed a statistically significant difference between TLHON-Kir and controls. One caveat is that unfortunately these events are not very common, particularly in controls (control flies tend to stay out of the heat altogether). Overall, even with many months of additional work and a much larger dataset, we do not think these effects are likely to become sufficiently convincing to be included in the paper.

3. DNp06 is the output neuron in this circuit. It is crucial to determine whether DNp06 is necessary and sufficient for temperature preference assays. Additionally, evidence for the functional connection between TLHONs and DNp06 is also important.

We apologize for not being more clear. We argue that DNp06 is a *relevant* output neuron in the TPN-III>TLHON circuit that mediates the response to a salient, rapid thermal change; DNp06 is unlikely to be *the sole* output neuron involved in heat escape or even altogether *the sole* output neuron involved in temperature preference behavior. When considering the complexity of multiple TPNs running in parallel, multiple 3Ns, the complexity of the circuit revealed by the EM connectome etc., our guess is that no single brain neuron will be “*necessary and sufficient*” for temperature preference.

When it comes to the specific aspect of behavior discussed here, for necessity, the manuscript’s **Figure 6** already shows that silencing DNp06 produces early turn phenotypes very similar to those produced by silencing TLHONs. For activation (sufficiency), optogenetic activation of DNp06 has already been published in two papers that we cite (PMID29943729, PMID36603587), and see above for our discussion of specific barriers that preclude us from performing more meaningful optogenetic experiments (i.e. in closed loop experiments).

We agree that it would have been good to strengthen the connectome data with functional connectivity experiments. The requested experiment requires recording GFP-expressing DNp06 by patch clamp, while as the same time activating TLHON e.g. by channelrhodopsin (as we did in **Figure 1** for TLHONs); we attempted to design this experiment, but unfortunately we encountered difficulties that we do not think we can resolve in a reasonable timeframe for this revision: (1) we attempted to produce a TLHON-LexA line to independently activate TLHON while targeting DNp06 for recording (by orthogonal labeling by GFP). We produced a TLHON-LexA construct and made transgenic flies, unfortunately this new construct did not reproduce the original expression pattern. (2) DNp06 is labeled by a split Gal4, requiring the recombination of transgenes in order to accomplish the complex genotype required (five transgenes in the same animal: two splitGal4 drivers and one each of the Lexa driver, UAS effector, LexAop effector). This process will require considerable time (and meiotic recombination between the constructs may or may not be possible depending on the position of the transgenes, some of which are not precisely mapped). (3) In pilot experiments, we found that DNP06-GFP

expression to be relatively weak which, combined with the position of its cell body, likely will make patching this cell quite difficult.

4. It is recommended that the authors adhere to the Drosophila gene nomenclature when referring to genotypes and mutants in both the text and figures of the manuscript.

Thank you for bringing this to our attention, note that we sometimes abbreviate genotypes in the main text for clarity, but full genotypes can always be found in the methods.

5. Figure 1:

(a) The red and brown colors in Figure 1a are difficult to differentiate. It would be beneficial to enhance the color contrast.

Thank you for this suggestion, we agree that color contrast is not optimal in the jpg version included in the manuscript, we'll make sure to keep an eye on the rendering of the contrast in the final figure during production.

(b) The quality of Figure 1b is suboptimal, making it challenging to discern the cell numbers depicted. Consider improving the image quality or providing additional details to help identify the cell numbers more accurately.

Thank you for this suggestion, we now mention cell numbers directly in the text.

(k) It is unclear why the authors employed a two-way ANOVA in Figure 1k. It is worth noting that similar figures, such as 2k and 3d, implemented a one-way ANOVA. Please provide a rationale for using different statistical analyses in these similar figures.

This was an oversight, thank you for pointing this out. We use 2-way ANOVA to test for statistically significant interaction between the Gal4 or LexA driver and the UAS or LexAop effector (Figure 1d, 2k,l, Figure 3d,e,g, Supplemental Figure 3d-g). We also submitted a table of statistics that lists the specific tests, p values, Ns and other relevant parameters for each data panel.

(k and l) In Figure 1k, the red peak 1 appears smaller than 25 Hz, whereas, in Figure 1l, the red peak 1 appears to be approximately 25 Hz. However, according to the legend, these two values should be the same. Please clarify and explain the discrepancy.

Because of the nature of the experiment (due to small variations in flow likely due to turbulence) every stimulus/response trace is not perfectly temporally aligned to every other one. As a result, the net effect of overlaying and averaging as we did in k is to produce some smoothing of the data. The panels in l are produced using the same traces but with no such smoothing (stimulus/response traces from different sweeps are not aligned to one another, simply the max stimulus and responses are extracted from each) and quantifications are therefore more accurate. Note that our next figures show a dependency of TPN-III firing on rate of thermal change (not yet discussed at this point in the paper), so the jitter seen in the responses is likely to be due to small differences in the rate of thermal change (again likely due to differences in flow).

(g) It would be beneficial to include a representative figure of CC>Chrimson in Figure 1g to provide a more comprehensive visualization of the experimental results.

As requested, we now show a trace for CC>Chrimson (**Revised Figure 1g**)

6. Figure 2: (e) Please provide additional information about Figure 2e in the legend to enhance understanding of the data presented. (g) To clearly indicate the presence of two cell bodies in Figure 2g, consider using arrows or arrowheads to highlight their locations. (h) The red and green color choices in Figure 2h may pose challenges for color-blind readers. It is advisable to select alternative colors. (j) The labeling of Gal4>P2X2 and Gal4/+ in Figure 2j may be confusing and difficult to understand. It would be helpful if the authors utilized the same labeling as in Figure 2h for consistency and clarity. Additionally, in the legend, if "synapse counts" and the "aggregate # of synapses" refer to the same concept, it is recommended to use consistent terminology throughout the legend.

Thank you for this suggestion, we have now revised the figure and legend following the reviewer's suggestions. We note that for Figure 2g we run out of colors: we prefer to keep GFP green for obvious reasons, but we feel that the usual purple palette which is standard for these 2-color images will create confusion with the purple used in Figure 2c.

7. Figures 3d and 2k: How did the authors explain the different results for the two-choice assay between 25 and 35°C in these two figures?

Our take is that overall the phenotype is as similar as it could be expected when considering that in the 4-field chamber there is a pretty robust air flow that is absent in our more traditional 2-choice assays. We know that an antenna-independent pathway contributes to the avoidance of 35°C and 40°C (see Simoes et al., 2021), perhaps transduction through this pathway is potentiated by the presence of air flow.

8. Figure 3g: The labeling is not clear. It is recommended to change the labeling of experimental groups to dark grey.

Thank you for this suggestion, we think the reviewer is referring to the blue, brown and green boxes in **g**. We use these colors to highlight the olfactory channels that directly target TLHONs, as in **b**. We edited the legend to make this clear.

9. Figure 4j: In the top panel, depolarization is observed. It would be beneficial for the authors to provide an example demonstrating that when the temperature change rate is lower than 0.2°C/s, there is no activation observed.

Indeed, this figure shows that some depolarization is present but that this does not lead to spiking. This is consistent with our model whereby TLHON spiking responses have a threshold despite a more linear TPN-III input (for slower stimuli TPN-III input does not bring TLHONs to threshold).

10. Figure 5c, 5g, 7e: The statistical analysis of 25°C vs 40°C is not discussed in the figure legends. Additionally, in Figures 5g and 7e, it is necessary to label the temperatures of each assay (e.g., 25°C vs 30°C, 25°C vs 35°C, and 25°C vs 40°C).

Thank you for pointing this out. We better connected 5g with the neighboring labels to make the experimental conditions clear. We added labels to 7d, and we have now revised the legend with p values and tests.

11. In Figure 5d, the authors stated that there "are not significantly different between control and experimental groups." However, the specific statistical method used in this analysis is not mentioned.

Thank you for this suggestion, we now mention the test in the legend.

REVIEWERS' COMMENTS

Reviewer #1 (Remarks to the Author):

The authors have responded to my comments with new data, analyses and discussions. I have no further major comments or criticisms. Here're some minor comments.

- Line#751: was → were
- Line#751: asterisk = $p < 0.05$. To → an asterisk means $p < 0.05$. To
- Line#771: test → text
- Line#776: the corresponding authors → the corresponding author

Reviewer #2 (Remarks to the Author):

The authors have throughly addressed all three reviewers' comments in the response letter and in the revised manuscript. I am satisfied that the new version is improved and will be a significant contribution to the field.

Reviewer #3 (Remarks to the Author):

I am happy with the revised work/explanation.